# Diversity in sea buckthorn (*Hippophae rhamnoides* L.) accessions with different origins based on morphological characteristics, oil traits, and microsatellite markers

He Li[1,2], Chengjiang Ruan[2]*, Jian Ding[2], Jingbin Li[2], Li Wang[2], Xingjun Tian[1,3]*

**1** School of Life Science, Nanjing University, Nanjing, P.R. China, **2** Key Laboratory of Biotechnology and Bioresources Utilization, Dalian Minzu University, Dalian, P.R. China, **3** Co-Innovation Center for Sustainable Forestry in Southern China, Nanjing Forestry University, Nanjing, P.R. China

* ruan@dlnu.edu.cn (CR); tianxj@nju.edu.cn (XT)

**Data Availability Statement:** All relevant data are within the manuscript and its Supporting Information files.

## Abstract

Sea buckthorn (*Hippophae rhamnoides*) is an ecologically and economically important species. Here, we assessed the diversity of 78 accessions cultivated in northern China using 8 agronomic characteristics, oil traits (including oil content and fatty acid composition) in seeds and fruit pulp, and SSR markers at 23 loci. The 78 accessions included 52 from ssp. *mongolica*, 6 from ssp. *sinensis*, and 20 hybrids. To assess the phenotypic diversity of these accessions, 8 agronomic fruit traits were recorded and analyzed using principal component analysis (PCA). The first two PCs accounted for approximately 78% of the variation among accessions. The oil contents were higher in pulp (3.46–38.56%) than in seeds (3.88–8.82%), especially in ssp. *mongolica* accessions. The polyunsaturated fatty acid (PUFA) ratio was slightly lower in the seed oil of hybrids (76.06%) than that of in ssp. *mongolica* (77.66%) and higher than that of in ssp. *sinensis* (72.22%). The monounsaturated fatty acid (MUFA) ratio in the pulp oil of ssp. *sinensis* (57.00%) was highest, and that in ssp. *mongolica* (51.00%) was equal to the ratio in the hybrids (51.20%). Using canonical correspondence analysis (CCA), we examined the correlation between agronomic traits and oil characteristics in pulp and seeds. Oil traits in pulp from different origins were correlated with morphological groupings ($r = 0.8725$, $p = 0.0000$). To assess the genotypic diversity, 23 SSR markers (including 17 loci previously reported) were used among the 78 accessions with 59 polymorphic amplified fragments obtained and an average PIC value of 0.2845. All accessions were classified into two groups based on the UPGMA method. The accessions of ssp. *sinensis* and ssp. *mongolica* were genetically distant. The hybrid accessions were close to ssp. *mongolica* accessions. The 8 agronomic traits, oil characteristics in seed and pulp oils, and 23 SSR markers successfully distinguished the 78 accessions. These results will be valuable for cultivar identification and genetic diversity analysis in cultivated sea buckthorn.

**Funding:** This research was financially supported by the Natural Science Foundation of China (NSFC) (Grant No. 31100489), which was received by He Li. https://isisn.nsfc.gov.cn/egrantweb/

**Competing interests:** The authors have declared that no competing interests exist.

## Introduction

Sea buckthorn (*Hippophae rhamnoides* L.) is a hardy winter shrub that is naturally distributed throughout Asia and Europe. It is an economically valuable species, divided into eight subspecies. Of these subspecies, ssp. *sinensis* and *mongolica* are mainly distributed in Asia, where they are abundant and commercially cultivated [1–2]. The fruits of sea buckthorn are rich in a variety of phytochemicals with physiological properties, such as lipids, carotenoids, ascorbic acid, tocopherols, and flavonoids [3–5]. The main applications for the fruits include food, cosmetics, and pharmaceutical products [6–7]. One of the most requested products for therapeutic practices is sea buckthorn oil, which is extracted from both seeds and fruit pulp. The applications of sea buckthorn oil include healing of the skin, mucosa, and immune systems, especially in cancer and cardiovascular disease therapy [8–9].

Two important parameters in analyzing oil yield and quality are oil content and fatty acid (FA) composition (referred to here as 'oil traits' for simplicity). Sea buckthorn seed and pulp oils are considered the most valuable products of the berries with a unique FA composition [10]. The seed oil contains omega-3 ($\alpha$-linolenic acid) and omega-6 (linoleic acid) FAs, and the pulp oil is characterized by a high concentration of FAs from the omega-7 group (e.g., palmitoleic acid). Seed oil is rich in unsaturated FAs (commonly 30–40% linoleic acid and 20–35% linolenic acid) [10]. The soft parts (pulp and peel) of the berries have an FA composition that differs from the seeds that is characterized by a high level of palmitoleic acid (16–54%), which is very uncommon in plants. The oil traits of sea buckthorn berries vary greatly according to their origin, based on the climatic and geological conditions of the growing areas [11].

Sea buckthorn adapts well to extreme conditions, including drought, salinity, alkalinity, and extreme temperatures [12]. Its vigorous vegetative reproduction and strong, complex root system with nitrogen-fixing nodules make it an optimal pioneer plant for soil and water conservation. For these reasons, sea buckthorn is cultivated widely in arid and semiarid areas of China [13]. However, the fruits of native cultivars are small and thorny and of low economic value, which encourages the breeding of sea buckthorn has undergone different stages of development in China, such as introduction, domestication, seedling selection and artificial hybridization for elite accessions. The cultivars of ssp. *mongolica* (introduced from Russia and Mongolia), ssp. *sinensis* (China origin) and hybrids (ssp. *mongolica* × ssp. *sinensis*) are abundant in northern China [14]. However, as a perennial woody plant, traditional cross breeding that takes long time and has low efficiency cannot meet the needs of modern production in sea buckthorn. It is essential for economic production to utilize molecular marker-assisted breeding (MAB) in sea buckthorn, especially to breed accessions with desirable oil traits. An essential step in this process is the genetic analysis of sea buckthorn germplasm. At present, molecular markers are mainly used for the analysis of genetic diversity, the taxonomic and geographic origin of cultivars, sex determination and population genetic structure in sea buckthorn [14–16]. SSR (simple sequence repeat, microsatellite) markers, with 1- to 6-bp DNA regions repeated in tandem, have been used in these analyses for their advantages of codominance, random distribution throughout the genome, easy detection, and high polymorphism and reproducibility [17]. Currently, an increasing number of microsatellite markers are being developed in sea buckthorn using high-throughput sequencing techniques for transcriptome datasets (RNA-Seq), which have become valuable resources for SSR discovery [14, 18]. In our previous study, 17 RNA-Seq SSR markers (SB1-SB17) were developed and validated on 31 accessions, which were utilized in the present study for genetic diversity assessment of larger set of accessions [14].

Diversity analysis helps clarify the relationships between germplasm characteristics and genotype and will improve our understanding of sea buckthorn germplasm to achieve greater production with higher quality regarding the important traits correlated with germplasm [19].

In the present study, 78 accessions of sea buckthorn with variation in fruit traits were selected as materials. The aim of this study is to report the phenotypic characteristics and oil traits in fruit pulp and seeds and the genetic diversity of the 78 sea buckthorn accessions in northern China, providing a foundation for MAB in sea buckthorn.

## Materials and methods

### Plant materials

Berries and leaves of 78 sea buckthorn accessions belonging to ssp. *mongolica* (52 accessions), ssp. *sinensis* (6 accessions) and hybrids (ssp. *mongolica* × ssp. *sinensis*, 20 accessions) were collected from the end of July to mid-September in 2015. Table 1 summarizes information on the plant materials. Three research institutes located in northern China, the Institute of Selection and Breeding of *Hippophae* (42°26′N, 121°28′E; 380 m) in Fuxin, the Research Institute of Berry (47°14′N, 127°06′E; 202 m) in Suiling and the Jiuchenggong Breeding Base of Sea Buckthorn (39°40′N, 110°09′E; 1400 m) in Dongsheng, provided 76 accessions of sea buckthorn samples (Fig 1, S1 Table). The other two accessions, Quyisike and Zhongguoshaji[wild], were harvested from cultivated fields in Qinghe (46°40′N, 90°22′E; 1218 m) and Datong (36°53′N, 101°35′E; 2800 m) (Fig 1, S1 Table, S2 Table). These areas have various geographical and climatic conditions (S3 Table).

The young leaves of each plant were kept at −80°C for use. The berries of each accession were pooled and frozen as quickly as possible at −20°C. When all plant materials were harvested, the berries were transferred to −50°C for storage until analysis.

### Morphological characteristics of fruit

Hundred berry weight (HBW) was the weight of 100 fresh berries after they were picked from bushes. Hundred seed weight (HSW) was the weight of 100 seeds after air drying at room temperature (25°C) for 2 weeks [20]. There were three biological replicates for each measurement. The transverse and longitudinal diameters of berries (BTD and BLD) and the length, width and thickness of seeds (SL, SW and ST) were measured over 20 times each (on average) by micrometer calipers. The berry shape indices (BSIs) were estimated by the ratio of BLD to BTD. The minimum (Min), maximum (Max), mean ± standard deviation (SD), and coefficient of variation (CV%) were reported.

### Oil extraction and FA analysis in seeds and pulp

The methods of lipid extraction, transesterification (methylation) and purification of methyl esters of the lipid extracts were described by Yang and Kallio [11]. Briefly, samples (1 g) of seeds and fruit pulp were isolated from freeze-dried berries and lipids from the samples were extracted with chloroform/methanol (2:1, v/v) with mechanical homogenization of the tissues. The purified oils were filtered before the solvent was removed on a rotary evaporator. The lipids were weighed, and the oil contents (percentages) in seeds and fruit pulp were calculated. Three biological replicates were taken for analysis. Lipids were stored in chloroform at −20°C until analysis.

The oil (10 mg) was transesterified by sodium methoxide catalysis [11, 21]. It was dissolved in sodium-dried diethyl ether (1 ml) and methyl acetate (20 μl). Then, 1 M sodium methoxide in dry methanol (20 μl) was added, and the solution was agitated briefly and incubated for 5 min at room temperature. The reaction was stopped by adding a saturated solution of oxalic acid in diethyl ether (30 μl) with brief agitation. The mixture was centrifuged at 1500 g for 2 min, and the supernatant was dried in a gentle stream of nitrogen. Fresh hexane (1 ml) was added and the solution was filtered with microporous filtering films (0.22 μm) for analysis.

**Table 1. Accessions of sea buckthorn used for the study.**

| No. | Accession name | Abbrev.[a] | Collection site | ssp.[b] | No. | Accession name | Abbrev.[a] | Collection site | ssp.[b] |
|---|---|---|---|---|---|---|---|---|---|
| 1 | Zhuangyuanhuang | ZYH | Fuxin | M | 40 | E13-10 | E13-10 | Suiling | M |
| 2 | Wucifeng | WCF | Fuxin | M | 41 | E13-11 | E13-11 | Suiling | M |
| 3 | Liusha-1 | LS1 | Fuxin | M | 42 | E13-14 | E13-14 | Suiling | M |
| 4 | Siberia rumianes | SR | Fuxin | M | 43 | HS-1 | HS1 | Suiling | M |
| 5 | Fangxiang | FX | Fuxin | M | 44 | HS-4 | HS4 | Suiling | M |
| 6 | Yalishanda-12 | YLSD12 | Fuxin | M | 45 | HS-9 | HS9 | Suiling | M |
| 7 | Jiuyuehuang | JYH | Fuxin | M | 46 | HS-10 | HS10 | Suiling | M |
| 8 | Nanren | NR | Fuxin | M | 47 | HS-12 | HS12 | Suiling | M |
| 9 | Botanical garden | BG | Fuxin | M | 48 | HS-14 | HS14 | Suiling | M |
| 10 | Zajiao-1 | ZJ1 | Fuxin | H | 49 | HS-18 | HS18 | Suiling | M |
| 11 | Zajiao-2 | ZJ2 | Fuxin | H | 50 | HS-20 | HS20 | Suiling | M |
| 12 | Zajiao-3 | ZJ3 | Fuxin | H | 51 | HS-22 | HS22 | Suiling | M |
| 13 | MZ-14 | MZ14 | Suiling | M | 52 | Xin'e-1 | XE1 | Suiling | M |
| 14 | Shoudu | SD | Suiling | M | 53 | Xin'e-2 | XE2 | Suiling | M |
| 15 | Fenlan | FL | Suiling | M | 54 | Xin'e-3 | XE3 | Suiling | M |
| 16 | Aertai | AET | Suiling | M | 55 | Zhongguoshaji | ZGSJ | Suiling | S |
| 17 | Chengse | CS | Suiling | M | 56 | EZ-4 | EZ4 | Suiling | H |
| 18 | Chuyi | CY | Suiling | M | 57 | Za-56 | Za56 | Suiling | H |
| 19 | Hunjin | HJ | Suiling | M | 58 | Za1-2 | Za1-2 | Suiling | H |
| 20 | Jinse | JS | Suiling | M | 59 | Za05-6 | Za05-6 | Suiling | H |
| 21 | Juren | JR | Suiling | M | 60 | Za05-20 | Za05-20 | Suiling | H |
| 22 | Xiangyang | XY | Suiling | M | 61 | Za05-21 | Za05-21 | Suiling | H |
| 23 | Yousheng | YS | Suiling | M | 62 | Za4 | Za4 | Suiling | H |
| 24 | Katuni | KTN | Suiling | M | 63 | Za13-19 | Za13-19 | Suiling | H |
| 25 | Wulangemu | WLGM | Suiling | M | 64 | Za13-25 | Za13-25 | Suiling | H |
| 26 | TF1 | TF1 | Suiling | M | 65 | Juda | JD | Dongsheng | S |
| 27 | TF2-13 | TF2-13 | Suiling | M | 66 | Jianpingdahuang | JPDH | Dongsheng | S |
| 28 | TF2-23 | TF2-23 | Suiling | M | 67 | Manhanci | MHC | Dongsheng | S |
| 29 | TF2-24 | TF2-24 | Suiling | M | 68 | Zhongxiongyou | ZXY | Dongsheng | S |
| 30 | TF2-36 | TF2- 36 | Suiling | M | 69 | Liaofuza | LFZ | Dongsheng | H |
| 31 | Suiji-1 | SJ1 | Suiling | M | 70 | Zaciyou-1 | ZCY1 | Dongsheng | H |
| 32 | Suiji-3 | SJ3 | Suiling | M | 71 | Zaciyou-10 | ZCY10 | Dongsheng | H |
| 33 | Suiji-4 | SJ4 | Suiling | M | 72 | Zaciyou-12 | ZCY12 | Dongsheng | H |
| 34 | HD-3 | HD3 | Suiling | M | 73 | Xinzaci-26 | XZC26 | Dongsheng | H |
| 35 | E10-06 | E10-06 | Suiling | M | 74 | Shiciyou-2 | SCY2 | Dongsheng | H |
| 36 | E10-34 | E10-34 | Suiling | M | 75 | Shiciyou-5 | SCY5 | Dongsheng | H |
| 37 | E10-42 | E10-42 | Suiling | M | 76 | Shiciyou-30 | SCY30 | Dongsheng | H |
| 38 | E10-47 | E10-47 | Suiling | M | 77 | Zhongguoshaji[wild] | ZGSJ[wild] | Datong | S |
| 39 | E13-00 | E13-00 | Suiling | M | 78 | Qiuyisike | QYSK | Qinghe | M |

[a] Abbrev., abbreviation.

[b] ssp., subspecies; M, ssp. *mongolica*; S, ssp. *sinensis*; H, hybrid (ssp. *mongolica* ♀ × ssp. *sinensis* ♂).

Fatty acid methyl esters (FAMEs) were analyzed with a gas chromatography-tandem mass spectrometry (GC/MS/MS) system (model AxION® iQT™, PerkinElmer, Shelton, CT, USA). Chromatographic separation was achieved using a DB-23 capillary column (60 m × 0.25 mm × 0.25 μm; Agilent Technologies, Santa Clara, CA, USA) with the following temperature

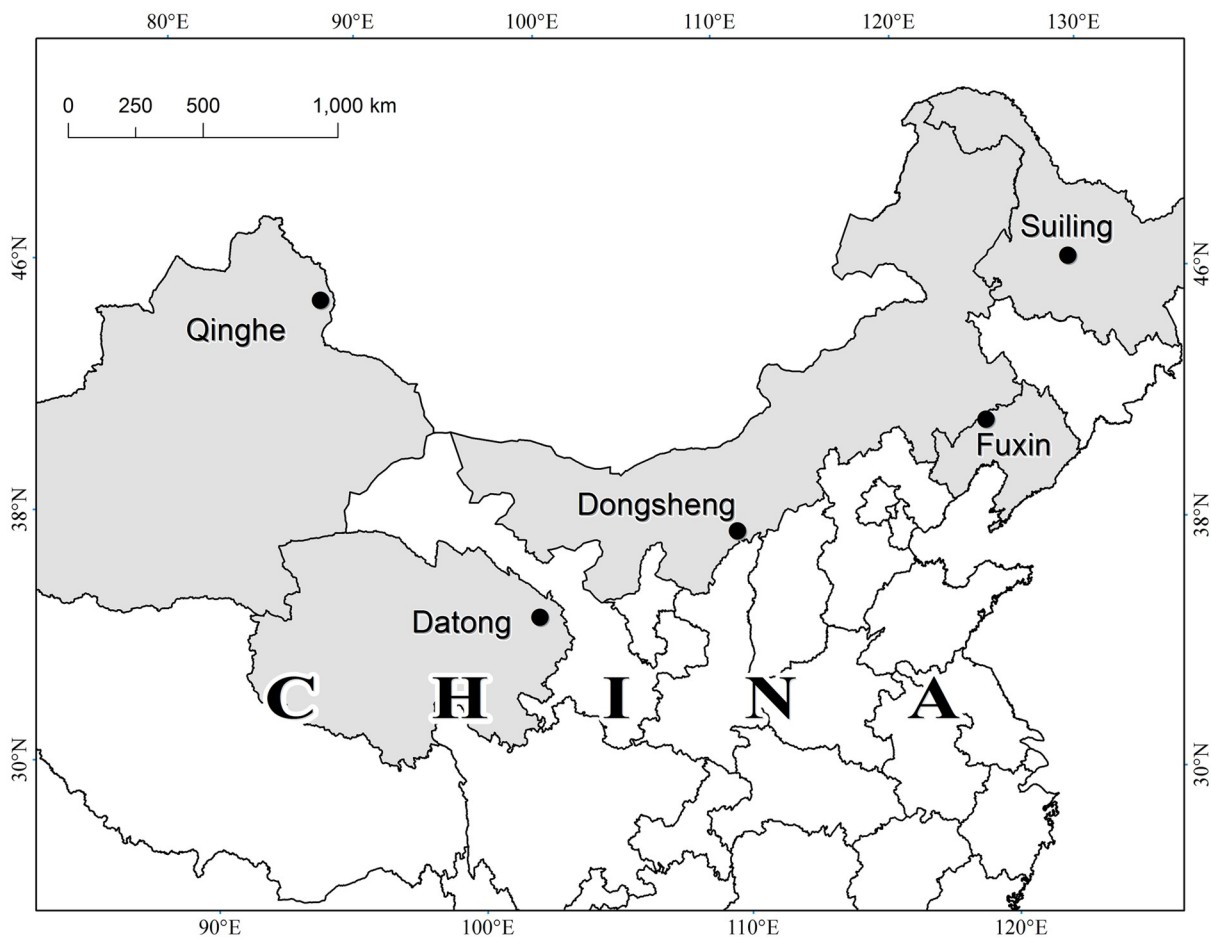

**Fig 1. The 78 sea buckthorn accessions from five cultivated lands used in this study.**

program: initial temperature 50˚C, hold for 1 min, heat to 175˚C at 25˚C/min, then heat to 215˚C at 3˚C/min and hold for 10 min, heat to 230˚C at 3˚C/min and hold for 5 min. The inlet was operated in split mode (1:20) at a temperature of 250˚C with helium as the carrier gas at constant flow of 1.0 ml/min. The transfer line temperature was 215˚C, and the MS ion source was set to 230˚C. MS detection was carried out in electron impact (EI) ionization mode, scanning all masses from 45–400 amu. FAME components were identified based on mass spectral comparison with an external standard (Supelco 37 Component FAME Mix, Sigma-Aldrich, St. Louis, MO, USA) and previous studies [10–11]. The main FA composition was expressed as a weight percentage of the total FAs from three replicates. The minimum, maximum, mean ± SD, and CV% were reported.

## Statistical analysis

The data analysis for morphological traits and oil characteristics was performed with SPSS[®] 24.0 (IBM[®]). The following parameters were evaluated: mean, minimum value, maximum value, SD and CV%. One-way analysis of variance (ANOVA) was used in the comparison of all traits among subsp. of *sinensis*, subsp. of *mongolica* and hybrids. Pearson correlation coefficients were calculated to analyze the relationship between pairs of 8 agronomic traits. Principal component analysis (PCA) was used to determine relationships among the accessions. In

addition, a canonical correspondence analysis (CCA) was applied to the data between morphological characteristics and oil traits in different tissues (seeds and pulp).

## DNA extraction and SSR analysis

Total genomic DNA was extracted from young leaves using the TaKaRa MiniBEST Plant Genomic DNA Extraction Kit (TaKaRa, Beijing, China) based on the manufacturer's protocol. The purity and quantity of extracted DNA were evaluated by gel electrophoresis and a Nano-Drop 2000 Spectrophotometer (Thermo Fisher Scientific, Waltham, MA, USA). Twenty-three polymorphic microsatellite loci (SB1-SB23) developed using RNA-Seq were evaluated. Of these, 17 (SB1-SB17) had been deployed in a previous study by the group [14] (S4 Table). PCR amplification was performed in 20 μl volumes containing 40 ng of DNA template, 1× PCR buffer, 1.5 mM $MgCl_2$, 0.15 mM of each dNTP (Takara, Dalian, China), 1.5 U of Taq polymerase (Takara, Dalian, China) and 0.5 μM of each primer. The PCR conditions included an initial denaturation at 94˚C for 2 min and 35 cycles of 30 s at 94˚C for denaturation, 30 s at 54–60˚C for annealing and 45 s at 72˚C for extension, with a final extension for 7 min at 72˚C using a C1000 Touch™ Thermal Cycler (Bio-Rad, Berkeley, CA, USA). PCR products were electrophoresed on 8% nondenaturing polyacrylamide gels using an SE 600 Ruby Standard Dual Cooled Vertical Unit (GE Healthcare Life Sciences, Pittsburgh, PA, USA) and visualized by silver staining.

The microsatellites were scored as codominant markers for genetic diversity analysis. The number of alleles (Na), effective number of alleles (Ne), observed and expected heterozygosity (Ho and He), Shannon's information index (Is) and polymorphic information content (PIC) for each of the genic SSR markers were calculated using GenAlEx 6.5 [22–23] and PowerMarker version 3.25 [24] software packages. A genetic similarity matrix based on the proportion of shared alleles was generated, and a UPGMA tree was constructed using PowerMarker. The dendrogram was displayed using MEGA 6 software [25] to reveal genetic relationships between the 78 sea buckthorn accessions.

## Results

### Morphological characterization of berries and seeds

Descriptive statistical analysis of 8 agronomic fruit traits for the 78 sea buckthorn accessions is shown in Table 2, S5 and S6 Tables. Relatively high CV values were observed for the HBW, BLD, and HSW (> 20%). The highest CV% was observed for the HBW (39.12%), which varied

**Table 2. Fruit traits of sea buckthorn berries of two different subspecies and hybrid accessions[a].**

| Trait name | Abbrev.[b] | ssp. *mongolica* | ssp. *sinensis* | Hybrid |
|---|---|---|---|---|
| Hundred berry weight (g) | HBW (g) | 47.69 ±11.03a | 10.73 ± 1.54c | 31.44 ±13.84b |
| Berry transverse diameter (mm) | BTD (mm) | 8.17 ± 0.99a | 5.84 ± 0.23b | 7.61 ± 1.24a |
| Berry longitudinal diameter (mm) | BLD (mm) | 10.90 ± 1.48a | 5.20 ± 0.19c | 8.15 ± 1.18b |
| Berry shape index | BSI | 1.35 ± 0.20 | 0.90 ± 0.05 | 1.08 ± 0.11 |
| Hundred seed weight (g) | HSW (g) | 1.60 ± 0.28a | 0.79 ± 0.23c | 1.28 ± 0.25b |
| Seed length (mm) | SL (mm) | 5.91 ± 0.68a | 3.31 ± 0.27c | 4.64 ± 0.56b |
| Seed width (mm) | SW (mm) | 2.76 ± 0.27a | 2.18 ± 0.18c | 2.52 ± 0.22b |
| Seed thickness (mm) | ST (mm) | 1.98 ±0.18a | 1.67 ± 0.16 b | 1.86 ± 0.26a |

[a] Values with different lowercase letters (a–c) are significantly different at $p < 0.05$.

[b] Abbrev., Abbreviation.

from 8.52 to 69.74 g. ANOVA ($p < 0.05$) showed that the HBW of ssp. *mongolica* berries was 47.69 ± 11.03 g, which was much higher than those of ssp. *sinensis* berries (10.73 ± 1.54 g) and hybrids (31.44 ± 13.84 g). In hybrids, the HBW values were high in EZ4, Za56, Za1-2, Za05-6 and Za05-21($> 45$ g), which were approximately the size of those in ssp. *mongolica* berries on average (S6 Table). The BTD varied from 5.54 to 10.80 mm, and the BLD varied from 4.83 to 14.25 mm. In addition, the BLD of berries from ssp. *mongolica* was higher than the BTD, and this relationship was the opposite in berries of ssp. *sinensis*. According to BSI values, the berry shapes of the three groups were significantly different ($p = 0.000$): oblong berries for ssp. *mongolica* (1.35 ± 0.20), oblate berries for ssp. *sinensis* (0.90 ± 0.05) and circular berries for the hybrids (1.08 ± 0.11). The HSW varied from 0.61 to 2.19 g with an average of 1.45 g. Similar to the HBW, there were significant differences in the HSW among seeds from ssp. *mongolica*, ssp. *sinensis*, and hybrids ($p = 0.000$). The SL varied from 2.00 to 3.49 mm, and the SW varied from 2.98 to 7.43 mm. The ST varied from 1.54 to 2.73 mm, with an average of 1.93 mm. Overall, the agronomic characteristics of seeds (HSW, SL, SW, and ST) showed relatively low coefficients of variation, ranging from 11.50–24.33%; however, the berries (HBW, BTD, BLD, and BSI) had high CV%s.

In previous multilocation trials in Suiling (47˚14′N, 127˚06′E; 202 m) and Dengkou (40˚ 43′N, 106˚30′E; 1053 m, Inner Mongolia), the fruit characteristics of 11 large berry accessions (AET, CS, CY, HJ, JS, JR, XY, YS, KTN, WLGM and SJ1) were comparatively analyzed (S7 Table). The HBWs values in Suiling (38.33–67.59 g) were higher than those in Dengkou (32.87–63.85 g). For all the introduced cultivars, the HBW values in the two experimental fields were lower than those in their country of origin, Russia. The phenotypic characteristics of sea buckthorn berries showed differences due to their origins, different parts of fruit analyzed, climatic and growing conditions. In this study, 78 accessions were selected for their good adaptabilities to growth sites.

PCA was performed using fruit characteristics (Fig 2). The first two PCs explained 78.11% of the total morphological variance. The first PC accounted for 41.74% of the variance. It was associated with BTD, HBW, ST, HSW, and SW in descending order. Therefore, these traits were important attributes for the classification of sea buckthorn accessions. The second PC accounted for 36.37%, which were correlated with BSI, SL, and BLD in descending order. The plot shows the distribution of 78 sea buckthorn accessions on PC1 and PC2 (Fig 2). The ssp. *mongolica* accessions with larger berries tended to cluster together, mainly positive on PC2. Six accessions of ssp. *sinensis* with the smallest berries were negative on both PC1 and PC2. The hybrids were largely distributed between the above two groups. Some hybrids (including ZCY1, ZCY10, ZCY12, XZC26, SCY2, and SCY5) were close to the accessions from ssp. *sinensis*.

## Oil characterization in seeds and seedless parts

The oil characteristics of seeds and seedless parts (pulp and peel) among the 78 accessions are summarized in Table 3 and Table 4. One special feature of sea buckthorn fruit was the high oil content in the pulp and peel (20.41%), in contrast to the oil content in the seeds (8.82%). A higher CV% was observed in pulp oil (42.72%) and varied over a wide range, from 3.46 to 38.56%. The pulp fraction of berries of ssp. *mongolica* had the highest oil content (24.68%) based on dry weight. The lowest pulp oil content (7.10%) on average was found in the berries of ssp. *sinensis*. In hybrids, the berries of ZJ2 contained 27.22% pulp oil, which slightly exceeded that of ssp. *mongolica* on average (S6 Table). The seed oil content varied from 3.88 to 12.75% with an average of 8.82%. The seeds of ssp. *mongolica* had the highest oil contents, with an average of 9.46%, and those of the other two groups did not differ significantly.

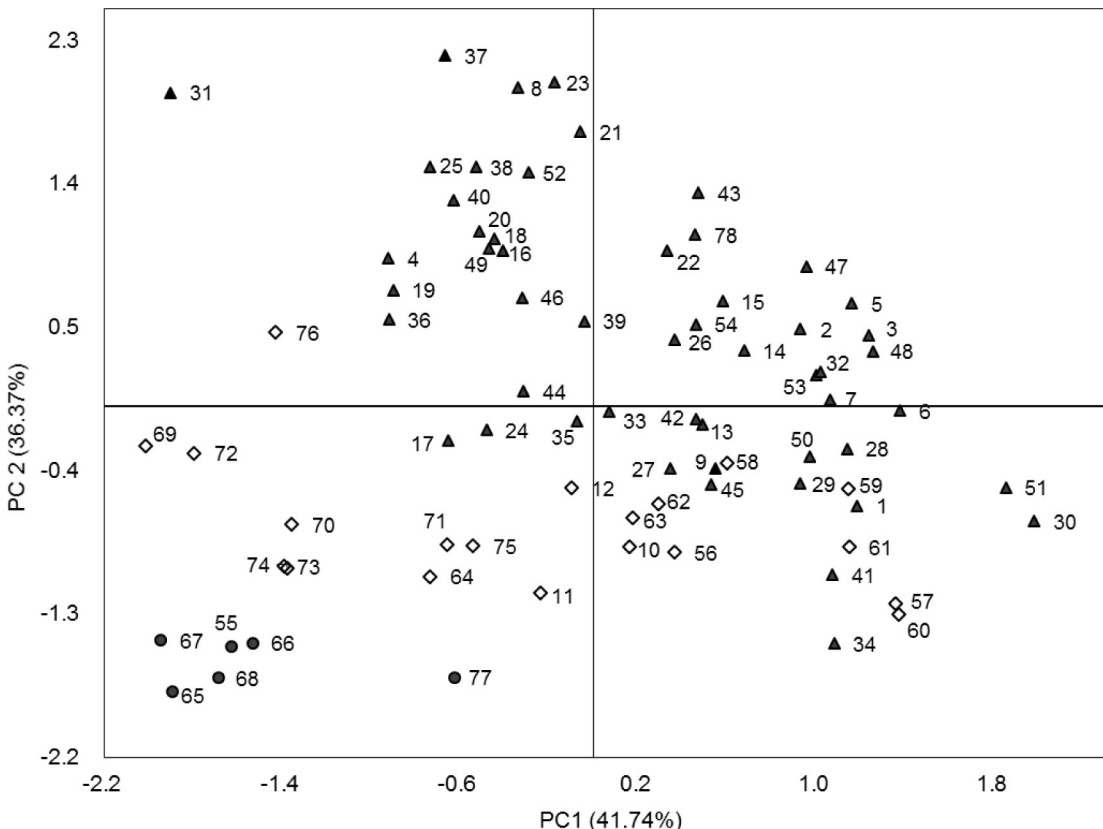

**Fig 2. Two-dimensional scatter plot for the first two principal components (PC1 and PC2) based on the agronomic fruit characteristics of 78 sea buckthorn accessions.** Numbers associated with symbols are the variety codes listed in Table 1. ▲ = ssp. *mongolica*; ● = ssp. *sinensis*; ◇ = hybrid.

For sea buckthorn, the FA composition in the seed oil differed significantly from that in the pulp oil. The proportions of FAs ranked from high to low as linoleic (18:2n6), α-linolenic (18:3n3), oleic (18:1n9), palmitic (16:0), stearic (18:0) and vaccenic (18:1n7) acids in the seed oil of most accessions (Table 4). Linoleic acid varied from 34.22 to 52.75% with an average of 42.17%. The proportion of α-linolenic acid varied from 21.37 to 47.16% with an average of 34.67%. High CV values were observed in oleic (30.50%) and vaccenic (39.17%) acids. Furthermore, the level of palmitoleic acid (16:1n7, < 0.5%) was extremely low in the seed oil. The FA composition of sea buckthorn seeds was similar among berries of the two different subspecies and hybrid accessions. Small variations were found in the proportion of linoleic acid in seed oil (40.44–42.87%). Its proportion in hybrids was slightly higher than in ssp. *mongolica* (42.87% vs 42.10%), and showed the highest mean value among the two different subspecies and hybrid accessions. α-Linolenic acid showed little variation, with a larger proportion in ssp. *mongolica* than in ssp. *sinensis* (35.56% vs 31.78%). A higher proportion of palmitic (7.41% vs 6.38%) and oleic (16.37% vs 13.96%) acids and a lower proportion of stearic acid (2.19% vs 2.23%) were discovered between the accessions of ssp. *sinensis* and hybrids. The polyunsaturated fatty acids (PUFA) ratio in hybrids (76.06%) was slightly lower than that in ssp. *mongolica* (77.66%) and higher than that in ssp. *sinensis* (72.22%). Some hybrids (including ZJ1, Za1-2, Za13-25, Za05-6, LFZ, and ZCY12) contained a high proportion of PUFAs (> 80%) in seed oil, which was more than the average level of ssp. *mongolica* accessions (S6 Table).

**Table 3. Oil characteristics of pulp and seeds of 78 sea buckthorn accessions (weight percentages).**

| Character | Pulp | | | | Seed | | | |
|---|---|---|---|---|---|---|---|---|
| | Min[a] | Max[b] | Mean ± SD[c] | CV[d] (%) | Min[a] | Max[b] | Mean ± SD[c] | CV[d](%) |
| oil content | 3.46 | 38.56 | 20.41 ± 8.72 | 42.72 | 3.88 | 12.75 | 8.82 ± 1.86 | 21.08 |
| Palmitic acid (16:0) | 24.52 | 53.08 | 36.26 ± 4.83 | 13.32 | 3.84 | 11.77 | 6.55 ± 1.39 | 21.16 |
| Palmitoleic acid (16:1n7) | 17.93 | 57.75 | 35.12 ± 7.64 | 21.76 | tr[e] | tr[e] | tr[e] | |
| Stearic acid (18:0) | 0.38 | 5.12 | 1.26 ± 0.70 | 55.58 | 1.41 | 4.58 | 2.16 ± 0.43 | 20.11 |
| Oleic acid (18:1n9) | 1.44 | 23.43 | 8.72 ± 4.72 | 54.13 | 3.05 | 25.95 | 13.25 ± 4.04 | 30.50 |
| Vaccenic acid (18:1n7) | 3.51 | 24.24 | 7.68 ± 4.09 | 53.28 | 0.45 | 2.38 | 1.20 ± 0.47 | 39.17 |
| Linoleic acid (18:2n6) | 3.02 | 17.40 | 9.97 ± 3.18 | 31.91 | 34.22 | 52.75 | 42.17 ± 3.60 | 8.54 |
| $\alpha$-Linolenic acid (18:3n3) | 0.12 | 7.16 | 1.00 ± 1.03 | 102.83 | 21.37 | 47.16 | 34.67 ± 4.42 | 12.75 |

[a] Minimum value.

[b] Maximum value.

[c] Standard deviation.

[d] Coefficient of variation expressed as a percentage.

[e] tr, trace ($< 0.5\%$).

In pulp oil, the dominant FAs were palmitoleic, palmitic, linoleic, oleic, and vaccenic acids (Table 3). Major differences were observed in the proportion of palmitoleic (17.93–57.75%), oleic (1.44–23.43%) and vaccenic (3.51–24.24%) acids. The special feature of pulp oil is high proportions ($> 35\%$) of palmitoleic and palmitic acids. Compared to ssp. *sinensis*, ssp. *mongolica* contained a higher proportion of palmitoleic and palmitic acids in the berry pulp ($p < 0.05$) (Table 4). In particular, the proportions of oleic and vaccenic acids were highest in ssp. *sinensis*, much higher than those in ssp. *mongolica* and hybrid accessions. The relative levels of $\alpha$-linolenic and stearic acids in pulp of ssp. *sinensis* were higher than ssp. *mongolica* ($p < 0.05$) (Table 4). For hybrids, the proportions of most FAs were between ssp. *mongolica* and ssp. *sinensis* accessions, except for linoleic acid. Similar to the results in seed oils, the hybrids had the highest proportions of linoleic acid (11.53%) and PUFA (12.60%). The monounsaturated fatty acid (MUFA) ratio in the pulp oil of ssp. *sinensis* (57.00%) was highest and that of ssp. *mongolica* (51.00%) was almost equal to that of the hybrids (51.20%). In the

**Table 4. Oil content and fatty acid composition in the seeds and fruit pulp of sea buckthorn berries of different origins[a] (weight percentages).**

| Character | Pulp oil | | | Seed oil | | |
|---|---|---|---|---|---|---|
| | ssp. *mongolica* | ssp. *sinensis* | Hybrid | ssp. *mongolica* | ssp. *sinensis* | Hybrid |
| oil content | 24.68 ± 6.79 a | 7.10 ± 3.28c | 13.34 ± 4.85b | 9.46 ± 1.56a | 6.70 ± 1.32b | 7.78 ±1.84b |
| Palmitic acid (16:0) | 37.68 ± 4.64a | 29.39 ± 3.71b | 34.62 ± 3.14a | 6.52 ± 1.16 | 7.41 ± 1.55 | 6.38 ± 1.82 |
| Palmitoleic acid (16:1n7) | 37.43 ±7.09a | 23.65 ± 4.16b | 32.55 ± 5.84a | tr[b] | tr[b] | tr[b] |
| Stearic acid (18:0) | 1.08 ±0.69b | 1.73 ± 0.64a | 1.59 ± 0.57ab | 2.13 ± 0.29 | 2.19 ± 0.44 | 2.23 ± 0.69 |
| Oleic acid (18:1n9) | 7.56 ±3.97b | 16.67 ± 6.84a | 9.33 ± 3.40b | 12.62 ± 3.75b | 16.37 ± 3.77a | 13.96 ± 4.46ab |
| Vaccenic acid (18:1n7) | 6.01 ±1.79c | 16.68 ± 6.20a | 9.32 ± 3.63b | 1.07 ± 0.37b | 1.80 ± 0.39a | 1.37 ± 0.55b |
| Linoleic acid (18:2n6) | 9.55 ±2.76ab | 8.34 ± 5.54b | 11.53 ± 2.92a | 42.10 ± 3.08 | 40.44 ± 4.06 | 42.87 ± 4.62 |
| $\alpha$-Linolenic acid (18:3n3) | 0.69 ±0.41b | 3.54 ± 2.09a | 1.07 ± 0.64b | 35.56 ± 4.13a | 31.78 ± 2.91b | 33.20 ± 4.89 ab |
| MUFA | 51.00 ±5.38b | 57.00 ± 9.46a | 51.20 ± 3.52b | 13.69 ± 3.93b | 18.18 ± 4.09a | 15.33 ± 4.90ab |
| PUFA | 10.24 ±2.98 | 11.89 ± 7.54 | 12.60 ±3.37 | 77.66 ± 4.31a | 72.22 ±5.54b | 76.06 ± 6.23ab |

[a] Values with different lowercase letters (a–c) are significantly different at $p < 0.05$.

[b] tr, trace ($< 0.5\%$).

hybrids, the pulp oil of SCY2 contained 39.16% palmitoleic acid, and the content of MUFAs was 60.77%, which was higher than that in ssp. *sinensis* (S6 Table).

## Correlations among the agronomic traits and oil characteristics

Canonical analyses allow direct comparisons of two data matrices. All sea buckthorn accessions were represented in a two-dimensional space using CCA between phenotypic traits and oil characteristics (Fig 3). For berries of the two different subspecies and hybrid accessions, phenotypic characters (BLD, HBW, BSI, and BTD) of berries and oil traits in pulp showed a close correlation ($r = 0.8725$, $p = 0.0000$). Based on CCA, accessions of ssp. *mongolica* were clustered on the upper side (mainly positive on D1 and D2), those of ssp. *sinensis* on the other, and the hybrids in the middle in Fig 3A. The positioning of samples in the first dimension was mostly related to differences in their berry characteristics that were primarily provided by the phenotypic character of BLD. The second dimension indicated differences in the oil contents and FA compositions of pulp oil among sea buckthorn accessions. Differences between pulp oil traits were primarily related to percentages of oil content, 16:0 and 16:1n7, which were highest in ssp. *mongolica*, followed by hybrids, and lowest in ssp. *sinensis*. For seeds of 78 accessions, phenotypic characteristics (SL, SW, ST, and HSW) and seed oil traits were correlated ($r = 0.7482$, $p = 0.0000$). The positioning of samples was staggered (Fig 3B), which reflected that all seed samples had relatively little variation among phenotypic traits and oil characteristics. These results verified the previous analysis (Table 2 and Table 3).

## SSR diversity

Twenty-three pairs of RNA-Seq SSR primers with good amplification and band stability were used among 78 accessions of sea buckthorn. A total of 69 bands were amplified using the 23 primer pairs, of which 59 were polymorphic, accounting for 85.51% of all bands. The number of amplified bands per locus ranged from 2 to 5, averaging 3, and Ne ranged from 1.0392 to 3.1049, averaging 1.6602 (Table 5). SB2, SB3, SB5, SB6, SB8, SB13, SB16 and SB23 were informative SSR loci, each revealing more than four effective alleles distributed among all of the accessions. Compared with Na, Ne and their average values were lower, which was caused by the uneven distribution of gene frequencies in SSR loci. In the genetic diversity analysis, Ho ranged from 0.0385 to 0.7949, with an average of 0.2965; He ranged from 0.0377 to 0.6779, with an average of 0.3291; and Is ranged from 0.0950 to 1.2152, with an average of 0.5681. The PIC value, regarded as discriminating power, varied from 0.0370 to 0.6174, with an average of 0.2845. Loci SB6 (PIC = 0.6174) and SB8 (PIC = 0.5820) showed higher effectiveness because of their high informativity and could be used to construct the fingerprint map of sea buckthorn germplasm. The characteristics of 23 polymorphic SSR markers in sea buckthorn accessions are shown in Table 5.

## Genetic relationships among sea buckthorn germplasm

Using 23 polymorphic SSR markers, the UPGMA dendrogram based on the proportion of shared alleles was constructed to assess the genetic relationships between the 78 accessions (Fig 4). The results showed that all the accessions could be divided into two groups (I and II). The accessions of ssp. *sinensis* (JD, ZGSJ, MHC, ZGSJ[wild], JPDH and ZXY) were clustered into group I. These accessions had closer relationships, despite great geographic differences. The second group was divided into 3 subgroups, namely, IIa, IIb, and IIc. The 20 hybrid accessions were all clustered into IIa. Subgroups IIb and IIc contained all the accessions of ssp. *mongolica* (introduced from Russia and Mongolia). Subgroup IIb included 6 accessions, namely WCF, LS1, QYSK, FX, SR, and MZ14. The remaining accessions of ssp. *mongolica* were clustered

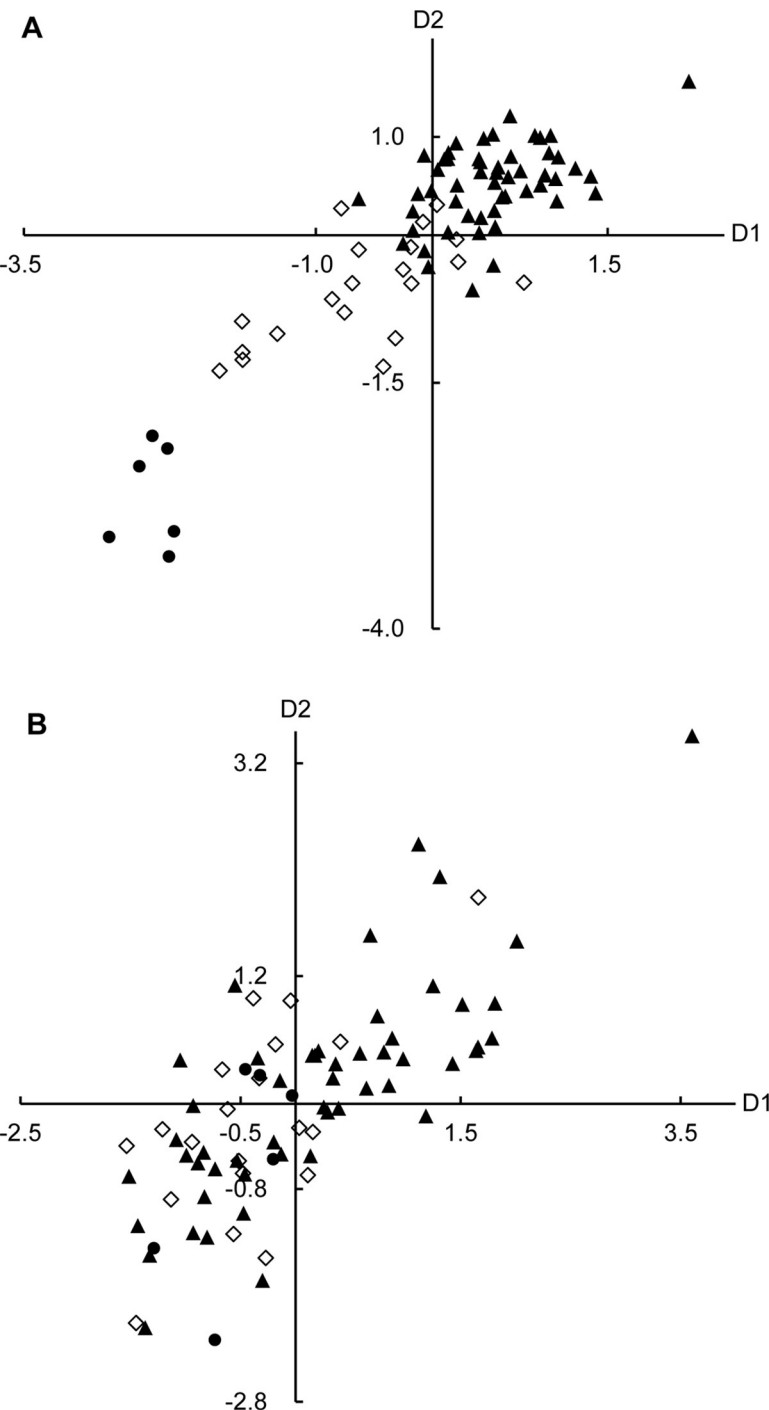

**Fig 3.** Canonical correspondence analysis of phenotypic traits (A. berry; B. seed) and oil characteristics (A. pulp oil; B. seed oil) of sea buckthorn germplasms. D1, Dimension 1; D2, Dimension 2. ▲ = ssp. *mongolica*; ● = ssp. *sinensis*; ◇ = hybrid.

into IIc. Among them, KTN, WLGM, HS4, HS9, HS10, HS12, HS14, HS18, HS20, HS22, WCF, FX and MZ14 composed one sub-subgroup. SJ3, ZYH, SD, NR, FL, XE2, XE3, JYH and YLSD12 showed close relationships. The other 23 accessions clustered into the third sub-

**Table 5. Genetic diversity analyses of 78 accessions of sea buckthorn germplasm using 23 SSR markers.**

| Loci code | Na | Ne | Ho | He | PIC | Is |
|---|---|---|---|---|---|---|
| SB1 | 3 | 1.2745 | 0.2436 | 0.2154 | 0.2025 | 0.3956 |
| SB2 | 4 | 1.1382 | 0.1282 | 0.1214 | 0.1166 | 0.2791 |
| SB3 | 4 | 2.2372 | 0.4615 | 0.5530 | 0.4627 | 0.9090 |
| SB4 | 2 | 1.5006 | 0.2692 | 0.3336 | 0.2779 | 0.5160 |
| SB5 | 4 | 2.1129 | 0.3333 | 0.5267 | 0.4735 | 0.9288 |
| SB6 | 4 | 3.1049 | 0.7051 | 0.6779 | 0.6174 | 1.2152 |
| SB7 | 2 | 1.0799 | 0.0769 | 0.0740 | 0.0712 | 0.1630 |
| SB8 | 5 | 2.8490 | 0.3846 | 0.6490 | 0.5820 | 1.1890 |
| SB9 | 2 | 1.1509 | 0.1410 | 0.1311 | 0.1225 | 0.2550 |
| SB10 | 3 | 1.5350 | 0.2949 | 0.3485 | 0.3114 | 0.6253 |
| SB11 | 2 | 1.9287 | 0.1667 | 0.4815 | 0.3656 | 0.6745 |
| SB12 | 3 | 1.2430 | 0.2179 | 0.1955 | 0.1753 | 0.3687 |
| SB13 | 4 | 2.1644 | 0.4231 | 0.5380 | 0.4392 | 0.8687 |
| SB14 | 2 | 1.9987 | 0.3077 | 0.4997 | 0.3750 | 0.6928 |
| SB15 | 2 | 1.0662 | 0.0641 | 0.0620 | 0.0601 | 0.1418 |
| SB16 | 4 | 1.4567 | 0.1923 | 0.3135 | 0.2956 | 0.6427 |
| SB17 | 2 | 1.4175 | 0.3590 | 0.2945 | 0.2512 | 0.4706 |
| SB18 | 2 | 1.0392 | 0.0385 | 0.0377 | 0.0370 | 0.0950 |
| SB19 | 3 | 1.0804 | 0.0641 | 0.0744 | 0.0724 | 0.1804 |
| SB20 | 2 | 1.1803 | 0.1667 | 0.1528 | 0.1411 | 0.2868 |
| SB21 | 3 | 1.9123 | 0.7308 | 0.4771 | 0.3802 | 0.7318 |
| SB22 | 3 | 1.2905 | 0.2564 | 0.2251 | 0.2025 | 0.4084 |
| SB23 | 4 | 2.4239 | 0.7949 | 0.5874 | 0.5102 | 1.0284 |

Na, observed number of alleles; Ne, effective number of alleles; Ho, observed heterozygosity; He, expected heterozygosity; PIC, polymorphism information content; Is, Shannon's information index.

subgroup. Overall, the relationship between ssp. *mongolica* and ssp. *sinensis* was relatively distant. The hybrids are close to ssp. *mongolica*, to which their female parents belonged.

## Discussion

Morphological characteristics, biochemical traits, and microsatellite markers have been used for germplasm identification and genetic diversity analysis in many horticultural plants [26–27]. The diversity at the morphological, biochemical, and molecular levels of 78 sea buckthorn accessions, composed of 52 from ssp. *mongolica*, 6 from ssp. *sinensis*, and 20 hybrids, was investigated.

The morphological characterization of plant materials with desired traits is an essential step for the effective use of germplasm [28]. Here, 8 important agronomic traits were measured among 78 sea buckthorn accessions, and a considerable amount of variation in morphological traits was found. The sizes of berries from the two different subspecies and hybrid accessions were significantly different according to the HBW value ($p$ = 0.000). Compared to ssp. *sinensis* berries, ssp. *mongolica* berries were much larger on average. The berry size of hybrid accessions was between the two subspecies. In the PCA, we plotted 2D plots with PC1 and PC2 scores of phenotypes (Fig 2). PC1 was mainly related to BTD and HBW, which explained the largest portion of the variance in 78 accessions. The distribution of 78 accessions on PC1 and PC2 was consistent with their agronomic characteristics (Fig 2). These results estimating morphological traits are valuable tools for identifying variation among plant germplasms [26].

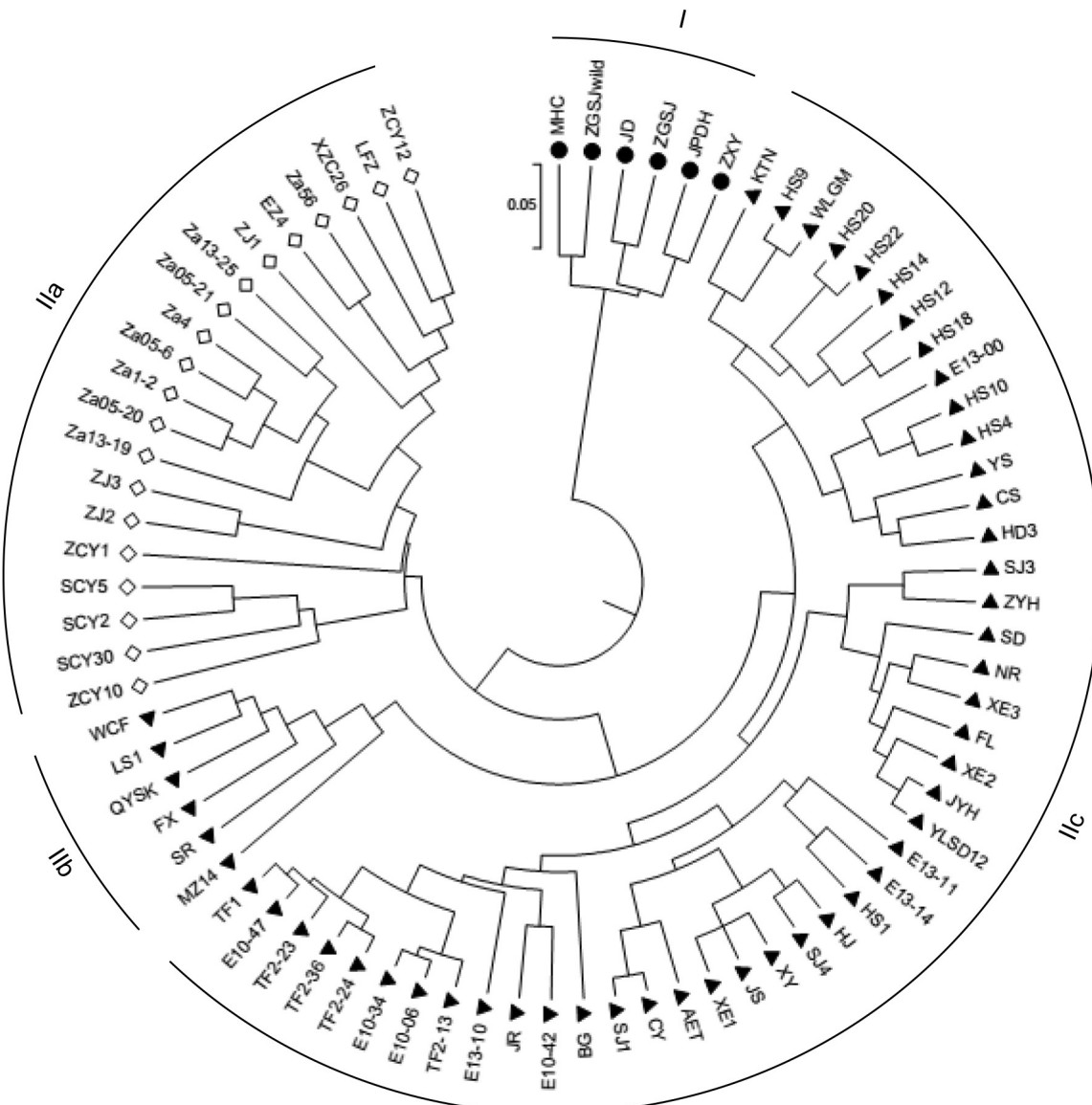

**Fig 4. UPGMA dendrogram of sea buckthorn germplasm based on SSR data (sample abbreviations described in Table 1).** ▲ = ssp. *mongolica*; ● = ssp. *sinensis*; ◇ = hybrid.

For biochemical traits, oil content and FA composition in the seeds and seedless parts were selected for their important roles in human health. The oil of sea buckthorn seems to be a good source of unsaturated FAs. Seed oil is rich in PUFAs, including linoleic and $\alpha$-linolenic acids. The proportion of PUFAs did not differ significantly among berries from three origins, despite the differences in some morphological characteristics and in growth conditions. These results were consistent with previous studies [10]. The results of the present study and previous investigations also suggested that the berries of ssp. *mongolica* were a good source of palmitic and palmitoleic acids in pulp oil and that those of ssp. *sinensis* were a good source of oleic acid in both seeds and fruit pulp [29]. Although carefully selected for intersubspecies crosses, some hybrids displayed elite oil traits. For example, the proportion of MUFAs in the pulp of SCY2 and of PUFAs in the seeds of 6 accessions (including ZJ1, Za1-2, Za13-25, Za05-6, LFZ, and

ZCY12) exceeded the average level of ssp. *mongolica* accessions, the subspecies that one of their parents belonged to. These results demonstrate the effectiveness of traditional cross breeding in the improvement of native accessions (ssp. *sinensis*), even though it is time-consuming and has low efficiency.

Previous studies found that berry size is a useful indicator of Vc, sugars and acids in population identification [19, 30]. The nutrients in the seedless fraction were more concentrated in the small berries of ssp. *sinensis* than in the large berries of ssp. *mongolica* [29]. In the present study, we analyzed the correlation between agronomic characteristics and oil traits at different levels (seed and pulp) by CCA. The results showed that the phenotypic characteristics (BLD, HBW, BSI, and BTD) of berries and the oil traits in pulp were positively correlated ($r = 0.8725$, $p = 0.0000$). The BLD, as a promising marker, provided the primary difference in CCA. Our results illustrated that berry size had different correlations with various biochemical characteristics in sea buckthorn.

Variation in phenotypic traits among germplasms may be attributed to differences in genetic backgrounds, geographical location, climate, harvest period and berry maturity, while molecular markers are independent of environmental condition and growth stage [31]. Twenty-three polymorphic SSR markers were used to investigate the genetic relationships among 78 sea buckthorn accessions. The 23 selected SSR markers detected 2–5 alleles, and their PIC values ranged from 0.1166 to 0.6155 and had an average of 0.3249. The PIC mean value was significantly lower than that of RAPD, ISSR and SRAP markers previously reported [15–16, 32], suggesting that the gene sequences of these SSR markers were conserved in sea buckthorn germplasm.

Based on UPGMA, the 78 accessions were classified into two groups. There is a large genetic distance between accessions of ssp. *sinensis* and ssp. *mongolica*. The hybrids were in between and rather close to ssp. *mongolica* accessions. Coincidentally, these hybrids were also between accessions of ssp. *sinensis* and ssp. *mongolica* on the PCA plot based on 8 agronomic characteristics. This uniformity indicated that the diversity of morphological characteristics could reflect genetic diversity and be used as markers in agronomy. Ruan et al. [15] assessed 14 Chinese, Russian and Mongolian sea buckthorn accessions using RAPD markers and obtained similar results. In a previous publication, the genetic relationship of 31 sea buckthorn accessions (also contained in this study) was analyzed based on 17 RNA-Seq SSRs [14]. However, the accessions of ssp. *mongolica* clustered in one group and those of ssp. *sinensis* and hybrids were divided in the other. This revealed that genetic relationships mainly relied on the diversity of genotypes and genetic backgrounds.

With the continuous development of high-throughput sequencing technology, transcriptome databases have become a powerful resource for SSR mining. An increasing number of RNA-Seq SSRs have been developed and applied to the study of species genetic diversity and population genetic structure [33–34]. The SSRs obtained by transcriptomes are associated with many important quantitative traits [35].

The results in the present study yielded useful knowledge regarding the diversity and genetic relationships of sea buckthorn germplasm in northern China, and could therefore facilitate further studies, including the selection of mapping populations and promising candidates, marker-trait association analysis based on establishing the consistency of the traits, and characterization of parents used in future breeding programs.

## Conclusion

In the present study, 8 phenotypic characteristics, oil traits in seeds and seedless parts, and 23 SSR markers successfully distinguished all 78 sea buckthorn accessions. In PCA, BTD and HBW in the first PC were the most important characteristics for distinguishing the accessions.

The agronomic traits of berries were closely correlated with the oil content and FA composition in the pulp by CCA. This information will be valuable for germplasm identification and genotypic diversity analysis in *H. rhamnoides*.

## Supporting information

**S1 Fig. 78 berry samples used in this study.** Numbers are the variety codes listed in Table 1.
(TIF)

**S2 Fig.** Total ion flow chromatography of 37 FAMEs Mix (A) and FAMEs in pulp oil in MHC (B).
(TIF)

**S1 Table. Samples of sea buckthorn grouped according to different genetic backgrounds.**
(DOCX)

**S2 Table. Characterization of the hybrids of sea buckthorn accessions studied.**
(DOCX)

**S3 Table. Climatic conditions at different growth sites of sea buckthorn samples in China.**
(DOCX)

**S4 Table. Primer sequences, annealing temperature, and estimated allelic size of 23 SSR markers.**
(DOCX)

**S5 Table. Descriptive statistics for morphological traits of berries and seeds among the sea buckthorn accessions studied.**
(DOCX)

**S6 Table. The morphological characteristics and oil traits of pulp and seeds of the 78 sea buckthorn accessions studied.**
(XLSX)

**S7 Table. Fruit traits and Vc contents of large berry accessions of sea buckthorn in two experimental fields (located in Suiling and Dengkou).**
(DOCX)

**S8 Table. Allele combinations obtained at the 23 microsatellite loci in 78 sea buckthorn accessions.**
(TXT)

## Acknowledgments

The authors are grateful to Hai Guo (Jiuchenggong Breeding Base of Sea Buckthorn) and Jun Zhang (Institute of Selection and Breeding of *Hippophae*) for the collection of plant materials.

## Author Contributions

**Conceptualization:** He Li, Chengjiang Ruan, Xingjun Tian.

**Data curation:** He Li, Chengjiang Ruan.

**Formal analysis:** He Li.

**Funding acquisition:** He Li.

**Investigation:** He Li.

**Methodology:** He Li, Chengjiang Ruan, Xingjun Tian.

**Project administration:** Chengjiang Ruan.

**Resources:** Jian Ding, Li Wang.

**Software:** He Li, Jingbin Li.

**Supervision:** Xingjun Tian.

**Writing – original draft:** He Li.

**Writing – review & editing:** Xingjun Tian.

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
