## [Decision Letter · Decision Letter 0]

23 Aug 2019

PONE-D-19-17567

Diversity in sea buckthorn (Hippophae rhamnoides L.) varieties with different origins based on morphological characteristics, oil traits, and microsatellite markers

PLOS ONE

Dear Dr. Tian

Thank you for submitting your manuscript to PLOS ONE. After careful consideration, we feel that it has merit but does not fully meet PLOS ONE’s publication criteria as it currently stands. Therefore, we invite you to submit a revised version of the manuscript that addresses the points raised during the review process.

The changes required for the revision might take longer then provided. If you need additional time, please let us know and we will extend the period for required revision. 

We would appreciate receiving your revised manuscript by 15 October, 2019. To enhance the reproducibility of your results, we recommend that if applicable you deposit your laboratory protocols in protocols.io, where a protocol can be assigned its own identifier (DOI) such that it can be cited independently in the future. For instructions see: http://journals.plos.org/plosone/s/submission-guidelines#loc-laboratory-protocols

We look forward to receiving your revised manuscript.

Kind regards,

Shailendra Goel, Ph.D.

Academic Editor

PLOS ONE

Journal Requirements:

Additional Editor Comments (if provided):

The manuscript “Diversity in sea buckthorn (Hippophae rhamnoides L.) varieties with different origins based on morphological characteristics, oil traits, and microsatellite markers” has been reviewed by two reviewers. It took me a while to secure reviewers for this manuscript, hence the delay, I apologies for it. Both the reviewers have submitted their comments, and one of them has also provided a sanitized version of the manuscript which can be seen by the authors. Both the reviewers have various concerns regarding the manuscript, but more importantly both have shown concern about the plant material and data generated. Since they raised a concern on the data, it will be better that manuscript is submitted along with answers to both the reviewer’s comments.

I will like to point out some important points:

1) The use of term varieties, cultivars, subspecies and hybrids have been without much explanation. For example, what is the basis of assigning hybrid status to a particular cultivar? More clarity is required in explanation of the material. How these varieties were assigned varietal status?

2) Generation of morphological dataset is also not mentioned clearly. You have 76 varieties growing at three locations. You need to provide environmental parameters for each location. Are all 76 growing at each location? If all the varieties are not growing at same location, many of morphological traits will be influenced by environmental factors. Did you do any multilocation trials to see the influence of environment on these traits? Did you try to collect data during different years and see if the data is consistent or showing variation. A statistical analysis of such data only will generate confidence in morphological data. Even a multilocation trial of a subset will provide information on reliability of data. Please include such data.

3) The sequencing data has been published earlier and 17 of SSR are coming from that data. Only 3 new markers have been used in the present study. This undermines the amount of data presented in this MS. You have to clearly mention these facts in the MS and the abstract. In my opinion more data needs to be generated. I suggest another 25-30 SSRs should be used for analysing the diversity.

4) There is no comparison given between the varieties used in previous publication and the present one. Are you using common varities? If you are than SSR data must be same and must have been presented in previous MS already. This has not been mentioned in the MS.

I hope you will appreciate comments made by two reviewers and will appreciate the time and efforts spent by them in reviewing the manuscript. The comments are positive and are an effort to improve the quality of manuscript further.

Reviewers' comments:

Reviewer's Responses to Questions

**Comments to the Author**

1. Is the manuscript technically sound, and do the data support the conclusions?

Reviewer #1: Partly

Reviewer #2: Partly

2. Has the statistical analysis been performed appropriately and rigorously? 

Reviewer #1: Yes

Reviewer #2: Yes

3. Have the authors made all data underlying the findings in their manuscript fully available?

Reviewer #1: Yes

Reviewer #2: No

4. Is the manuscript presented in an intelligible fashion and written in standard English?

Reviewer #1: No

Reviewer #2: Yes

5. Review Comments to the Author

Reviewer #1: General comments

The present study was conducted on 78 so called “varieties” of Hippophae rhamnoides, out of which two sets belonged to two sub-species and the third one is supposedly the hybrids. The study is aimed to decipher the genetic diversity and their oil content (in “berries”) of the “varieties”. With the given results and methodology authors are attempting to generated the data on two different aspects - first assessment of agronomic traits, especially the oil content and to relate it to a reliable phenotypic marker (among the eight selected morphological traits), and genetic diversity of species by using 20 SSR markers. An attempt has been made to link these traits together, which needs to be highlighted in the Introduction pertinently in the light of earlier attempts on related/unrelated taxa.

In general the MS needs to be modified for technical reasons and usage of sources in the study. A major revision of the MS is suggested before acceptance.

Specific comments

1. I think that in such bio-prospection studies sampling strategy is very crucial. The sampling method needs to explain that how these accessions were sourced. The MS needs elaboration on -

• How many individuals of a “variety” from each site were collected?

• Are these the random collections of registered varieties from the cultivated field in the five regions OR sampled from the wild?

• It is also not clear that how the hybrids were distinguished from parents while making collections.

• Do these sites differ in climatic conditions?

• What is the link of “origin” with oil content? Did you expect that there are bound to be differences because of differences in the climatic conditions of area of collection/cultivation of the same “variety/hybrid”?

Importantly instead of the term varieties the term accessions would be appropriate, as the authors have mentioned it in Table 1. According to the definition by The International Union for the Protection of New Varieties of Plants, "a variety must be recognizable by its characteristics, recognizably different from any other variety and remain unchanged through the process of propagation".

Do these two subspecies hybridize freely in nature and such hybrids have been characterized? This needs some population analysis like by using STRUCTURE, or at least there should be a note on the characterization of hybrids (including the features), even if they are procured form some Research Institute.

2. I don’t understand the usage of term pulp/peel in the MS (also see page 15, line 251). As the entire fleshy region was separately used for extraction of oil from the "berries" (see Methods), the use of term pulp would be appropriate. One cannot expect to remove the epidermal peel especially during the mechanical homogenization process.

3. How the present study for the genetic diversity analysis of 78 cultivars is different from other previous studies? May be highlighted in the introduction. Authors may also highlight that trait: i.e. Oil yield was correlated with the “promising” accessions.

4. Although attempt has been made of possible use of MAB in future, but it has not been justified with the discussion. For example, do the authors will depend on the same plants in the cultivated lands across the region or some mapping populations will be established. In former case GPS tagging of the individuals will be required for sourcing the material on regular basis and to establish the consistency of the trait.

Materials and Methods

5. Need to mention whether hundred-berry weight, hundred-seed weight and other dimensions were taken from mature or immature berries? In Supplementary figure 1 some samples are showing immature berries e.g. sample 65, 68 etc.

6. What do the ‘Berry Shape Indices’ refer to and what are its implications on the results/oil trait/ with genetic diversity. Provide any suitable reference if possible. (Page: 8, subsection: Morphological….)

7. The usage of phrase ‘8 agronomic traits’ seems to be superfluous as these are the traits of berries itself. How the seed width is different from the seed thickness? The difference is not apparent. Table 2 and 3; as well as in text.

8. The usage of abbreviation has not been followed see table 2 and 3. Table 2 is not necessary, may be omitted or shifted to Supplementary Data. In Tables SD is not mentioned.

9. The reference is missing for the SB18-SB20 SSRs; in the text (Page 10, line 181).

Results

10. Results should be given in the format mean ± SD. Minimum and maximum can be given in supplementary tables.

11. It is not clear from the table caption and content that whether values in the Table 4 is the minimum, maximum and mean values are representing the cumulative results of 78 varieties e.g. minimum in variety… and maximum in variety…. Need to mention in the results.

12. The results of CCA are driving a correlation between phenotypic traits and oil characteristics. The authors may use the information for total oil content (pulp+seed) or oil content in pulp and seeds separately for drawing any correlation. That would possibly help as a descriptor for the potential crop in identifying the elite/superior “variety” and further can be linked to genetic diversity.

Discussion

13. Page:28, Line:449-453. The link of this part of discussion is lacking with the previous text.

14. In conclusion part authors are concluding that this information may be useful for cultivar identification but initially they started their work for the varieties. Taxonomically these two are different entities.

Some suggestion:

1. The sequence of S1 and S2 table can be reversed as per the citation in the text.

2. Page:3, Line:54. Reference 1 is incorrect. The lead author here is Bartish I.V.

3. Page:3, Line:56-57. ….flavonoids [3-7]; ….products [8-10]. Here over-citation may be avoided.

4. Page:3, Line:59. ‘Sea buckthorn oil’ instead of ‘sea buckthorn oils’

5. Page: 4, Line 74. Add a reference to the statement. The plant is able to avoid cold and is not resistant, because the leaves are shed under extreme cold condition in this plant. Even the species is not resistant to alkali too.

6. Page:4. Line:85. Use full form at first place ‘MAB’.

7. Page:5. Line:110. What was the premise of including two known elite varieties in the study? Any supportive reference(s) for the statement, and also mention the context in which these varieties are elite.

8. Page:12. Line:204-205. May be included in Material and Methods.

Reviewer #2: The publication can be accepted post minor reviews.

Some of the comments to authors have been listed below. Some changes required are highlighted in the manuscript attached.

1. The authors mention that 76 varieties were used. There is no mention of the different species they belonged to in M&M, although it has been mentioned later in the text and table. Incorporate that information in the M&M.

2. Are these 76 different varieties or just different accessions? At many places they are being referred to as ‘cultivars’ also. Please correct accordingly in the text wherever mentioned.

3. How variable are the climatic conditions of the three research institutes?

4. Line 109: ‘………provided 76 varieties’. Does this mean that all the 76 were grown at all the 3 fields? There is no clarity on this aspect in the M&M. Most quantitative traits exhibit a huge variation across environments. To study the phenotypic variations it would have been much informative if all the 76 varieties were grown together across all the three fields. Why was that not considered?

5. There is no mention of how these varieties were grown in the field, and data from how many plants were considered for the morphological and oil analysis. For eg. for hundred berry weight (HBW), berries were collected from how many different plants?

6. Line 137: For the oil extraction and FA analysis, the authors mention that ‘each sample was analyzed three times’. Why weren’t three biological replicates taken for this analysis?

7. Line 180-181: The authors have used 17 previously developed SSR markers and 3 newly developed SSR markers using RNA-Seq. What was the basis of selection of just 3 new markers from the RNA-Seq. Why weren’t more markers deployed for the genetic characterization?

8. Line 180: Please reframe the sentence. It appears that the authors have done RNA-seq to generate the 3 new SSR markers. Although, the RNA-seq had been done in previous study from where the 17 SSR were also developed (Reference 17).

9. Instead of ‘different origins’ that has been used repeatedly by authors throughout the text and tables, I suggest use the two different species and hybrid accessions.

10. Line 340: ‘All the primers’. Reframe this line. All primers did not give 59 bands. A total of 59 bands were amplified.

11. Line 341: ‘accounting for 86.44%’ . Incomplete sentence, 86.44% of what??

12. Line 372: the 3 subgroups have been referred incorrectly. They are IIa, IIb and IIc.

13. Line 421: ‘in comparison of populations’. Statement not clear. Please reframe.

14. Line 436: ‘gene sequences’. Are all the SSR markers used genic in nature?

15. Table 1: Could just be described as the ‘Accessions of sea buckthorn used for the study’

6. PLOS authors have the option to publish the peer review history of their article (what does this mean?). If published, this will include your full peer review and any attached files.

Reviewer #1: No

Reviewer #2: No

---

## [Author Response · Author response to Decision Letter 0]

16 Nov 2019

Response to Reviewer' Comments letter to PLOS ONE

The authors thank the additional editor and two reviewers for their careful reading, comments and suggestion. We revised our manuscript in the best way as we could. Revised portions are marked in red in the revised manuscript. For the individual comments see our reply below.

Additional Editor Comments:

1) The use of term varieties, cultivars, subspecies and hybrids have been without much explanation. For example, what is the basis of assigning hybrid status to a particular cultivar? More clarity is required in explanation of the material. How these varieties were assigned varietal status?

Response: The Reviewer 1 gave the definition of ‘variety’ that "a variety must be recognizable by its characteristics, recognizably different from any other variety and remain unchanged through the process of propagation". The ‘cultivar’ refers to a variety of a plant developed from a natural species and maintained under cultivation. The authors accepted the reviewers’ advice that the term ‘accessions’ would be appropriate according to the plant materials in present study.

The hybrid accessions in this study generated by hybridization experiment in control between ssp. mongolica and ssp. sinensis at specialized experimental fields and selected for their desirable traits. After a complex process of identification of experts, some hybrids may became a new ‘cultivar’.

2) Generation of morphological dataset is also not mentioned clearly. You have 76 varieties growing at three locations. You need to provide environmental parameters for each location. Are all 76 growing at each location? If all the varieties are not growing at same location, many of morphological traits will be influenced by environmental factors. Did you do any multilocation trials to see the influence of environment on these traits? Did you try to collect data during different years and see if the data is consistent or showing variation. A statistical analysis of such data only will generate confidence in morphological data. Even a multilocation trial of a subset will provide information on reliability of data. Please include such data.

Response: The environmental parameters for each location were provided in the S2 Table of revised manuscript. All the accessions are not growing at same location. However, they could adapt to the environment of their cultivated lands well. We had performed some multi-location trials to see the influence of environment on berry characteristics before that was supplemented in the results of revised manuscript (S4 Table).

3) The sequencing data has been published earlier and 17 of SSR are coming from that data. Only 3 new markers have been used in the present study. This undermines the amount of data presented in this MS. You have to clearly mention these facts in the MS and the abstract. In my opinion more data needs to be generated. I suggest another 25-30 SSRs should be used for analysing the diversity.

Response: We have mentioned it in the MM and the abstract of the revised manuscript. We screened 3 new SSR loci (SB21-23) with polymorphism from 20 SSR primer pairs during the revision of the manuscript. These information has been supplemented in revised manuscript. It is difficult to develop more RNA-Seq SSRs. On one hand, the genic sequences used for developing SSR markers were highly conserved in sea buckthorn germplasm. On the other hand, the species and subspecies of sea buckthorn germplasm used in this study are limited to facilitate more polymorphism at SSR loci.

4) There is no comparison given between the varieties used in previous publication and the present one. Are you using common varities? If you are than SSR data must be same and must have been presented in previous MS already. This has not been mentioned in the MS.

Response: In previous publication, 31 accessions (common in the present one) were used for the validation of developed SSR markers. They included 6 accessions of ssp. sinensis, 14 accessions of ssp. mongolica and 11 hybrid accessions. They were selected according to their genetic origins and cultivated lands. In present study, the accessions were selected based on various fruit traits. The results of genetic relationship were different from that in the previous publication. That was supplemented in the discussion of revised manuscript. 

‘In previous publication, the genetic relationship of 31 sea buckthorn accessions (also contained in the 78 accessions) were analyzed based on 17 RNA-Seq SSRs [14]. However, the accessions of ssp. mongolica clustered in one group and those of ssp. sinensis and hybrid were in the other one. That revealed the genetic diversity is related on the genotypes and genetic backgrounds.’

Reviewer #1: 

Specific comments

1. I think that in such bio-prospection studies sampling strategy is very crucial. The sampling method needs to explain that how these accessions were sourced. The MS needs elaboration on –

• How many individuals of a “variety” from each site were collected?

• Are these the random collections of registered varieties from the cultivated field in the five regions OR sampled from the wild?

• It is also not clear that how the hybrids were distinguished from parents while making collections.

• Do these sites differ in climatic conditions?

• What is the link of “origin” with oil content? Did you expect that there are bound to be differences because of differences in the climatic conditions of area of collection/cultivation of the same “variety/hybrid”?

Importantly instead of the term varieties the term accessions would be appropriate, as the authors have mentioned it in Table 1. According to the definition by The International Union for the Protection of New Varieties of Plants, "a variety must be recognizable by its characteristics, recognizably different from any other variety and remain unchanged through the process of propagation".

Do these two subspecies hybridize freely in nature and such hybrids have been characterized? This needs some population analysis like by using STRUCTURE, or at least there should be a note on the characterization of hybrids (including the features), even if they are procured form some Research Institute.

Response: The part of ‘Plant materials’ in original text was revised according to above advice. 

• 235 individuals (2−5 ramet plants each accession) of 5−8 years in 5 growth sites were collected. 

• These are registered accessions from the cultivated field and adapt to local environment. 

• For the identification of the hybrid accessions, they are labelled and recorded with documents. Furthermore, most hybrid accessions and their parents are not in the same growth site. The parents of them are cultivated in the experimental field for hybridization. 

• The growth sites differ in climatic conditions which are described in S2 Table.

• According to the results in this study, the oil contents in pulp and seeds are highest in ssp. mongolica accessions on average. That is the link of origin with oil content. In this study, we ignored the difference in the climatic conditions of cultivated fields for the sea buckthorn accessions we selected adapted local environment well.

• The authors agreed the opinion that the term accessions would be appropriate and all the term varieties were revised to the term accessions.

• These two subspecies hybridized by experiments in control which were performed in specialized experimental fields, And the hybrid accessions are characterized in the Research Institutes. 

2. I don’t understand the usage of term pulp/peel in the MS (also see page 15, line 251). As the entire fleshy region was separately used for extraction of oil from the "berries" (see Methods), the use of term pulp would be appropriate. One cannot expect to remove the epidermal peel especially during the mechanical homogenization process.

Response: The authors accepted the advice and the phrase ‘pulp/peel’ in the original text was revised to ‘pulp’ in the revised manuscript.

3. How the present study for the genetic diversity analysis of 78 cultivars is different from other previous studies? May be highlighted in the introduction. Authors may also highlight that trait: i.e. Oil yield was correlated with the “promising” accessions.

Response: The related content has been supplemented in the introduction of the revised manuscript. 

‘The diversity analysis helps understand the relationships between germplasm characters and genotype will improve the sea buckthorn germplasm to achieve higher production of higher quality for the important traits were correlated with the promising germplasm [19]. 

In present study, 78 accessions of sea buckthorn with large variation of fruit traits were selected as materials.’

4. Although attempt has been made of possible use of MAB in future, but it has not been justified with the discussion. For example, do the authors will depend on the same plants in the cultivated lands across the region or some mapping populations will be established. In former case GPS tagging of the individuals will be required for sourcing the material on regular basis and to establish the consistency of the trait.

Response: The results in present study yielded useful knowledge regarding the diversity and genetic relationships of sea buckthorn germplasm in northern China, and could therefore facilitates further studies, including selection of mapping populations and promising candidates, marker-trait association analysis based on establishing the consistency of the traits , and characterizing parents used in future breeding programs. The above information on possible use of MAB in future has been supplemented in the discussion of the revised manuscript.

Materials and Methods

5. Need to mention whether hundred-berry weight, hundred-seed weight and other dimensions were taken from mature or immature berries? In Supplementary figure 1 some samples are showing immature berries e.g. sample 65, 68 etc.

Response: The hundred-berry weight, hundred-seed weight and other dimensions should be taken from mature berries. So the berries of all accessions were collected from the end of July to mid-September, according to their ripening stages. But it is difficult to collected ripening fruits of 78 sea buckthorn accessions. The berries of several accession were harvested when they are approaching maturity. So the data error existed in the dimensions of several accessions. The authors admitted it and hope be understood at this point.

6. What do the ‘Berry Shape Indices’ refer to and what are its implications on the results/oil trait/ with genetic diversity. Provide any suitable reference if possible. (Page: 8, subsection: Morphological….)

Response: The berry shape index (BSI) is estimated by the ratio of BLD to BTD, also called length/width ratio in some studies, which indicates berry shape. According to the results in present study, the phenotypic characters (BLD, HBW, BSI, and BTD) of berries and oil traits in pulp showed close correlation (r = 0.8725, p = 0.0000) using CCA. The relevant literature is below. The results of it showed that the morphological traits established were consistent with those derived from the SSR markers in olive plant materials. The length/width ratio was one of the morphological traits of endocarp in that study.

Patricia RR, Carmen GB, Beatriz CG, Jesús SG, Isabel T. Genotypic and phenotypic identification of olive cultivars from northwestern Spain and characterization of their extra virgin olive oils in terms of fatty acid composition and minor compounds. Sci Hort. 2018; 232:269-279.

7. The usage of phrase ‘8 agronomic traits’ seems to be superfluous as these are the traits of berries itself. How the seed width is different from the seed thickness? The difference is not apparent. Table 2 and 3; as well as in text.

Response: For sea buckthorn, the traits of berries (including seeds) are very important for their economic value. The seed thickness could be regarded as the ‘height’ of seeds, which is a parameter of oilseed, e.g. olive. 

8. The usage of abbreviation has not been followed see table 2 and 3. Table 2 is not necessary, may be omitted or shifted to Supplementary Data. In Tables SD is not mentioned.

Response: The authors accepted the advice. Table 2 was shifted to S4 Table. The data of ‘Mean 

9. The reference is missing for the SB18-SB20 SSRs; in the text (Page 10, line 181).

Response: The SB18-SB20 SSRs were firstly reported in this study and no reference could be given for them.

Results

10. Results should be given in the format mean ± SD. Minimum and maximum can be given in supplementary tables.

Response: The authors accepted the advice and the results have been given in the format mean ± SD. 

11. It is not clear from the table caption and content that whether values in the Table 4 is the minimum, maximum and mean values are representing the cumulative results of 78 varieties e.g. minimum in variety… and maximum in variety…. Need to mention in the results.

Response: The authors accepted the advice. The table caption in the Table 4 of original text is not clear because we want to use the abbreviation of ‘minimum, maximum’ but the notes were forgotten to give bellow the table. And these data have been mentioned in the results in the revised manuscript.

12. The results of CCA are driving a correlation between phenotypic traits and oil characteristics. The authors may use the information for total oil content (pulp+seed) or oil content in pulp and seeds separately for drawing any correlation. That would possibly help as a descriptor for the potential crop in identifying the elite/superior “variety” and further can be linked to genetic diversity.

Response: For the difference in the FAs composition between pulp oil and seed oil, the total oil content was not be used for drawing any correlation in this study. In practical production, the seed oil and pulp oil are separately extracted for their different functions. During the course of CCA, the factors in each data matrix would be analyzed by pairwise correlation analysis. So oil content in pulp and seeds separately for drawing any correlation is not necessary. 

Discussion

13. Page:28, Line:449-453. The link of this part of discussion is lacking with the previous text.

Response: In the part of ‘Introduction’, the superiority of SSR markers was mentioned. The significance of developing SSR markers with RNA-Seq technique was also mentioned in it. The SSR markers used in this study are developed by RNA-Seq. All these description was the link of this part of discussion.

14. In conclusion part authors are concluding that this information may be useful for cultivar identification but initially they started their work for the varieties. Taxonomically these two are different entities.

Response: The authors agreed this opinion. The phrase ‘cultivar identification’ was revised to ‘germplasm identification’ and all the word ‘varieties’ were changed into ‘accessions’ in the revised manuscript according to the taxonomical definition. 

Some suggestion:

1. The sequence of S1 and S2 table can be reversed as per the citation in the text.

Response: The good advice mentioned above is accepted by the authors. The tables were reversed in the revised manuscript. 

2. Page:3, Line:54. Reference 1 is incorrect. The lead author here is Bartish I.V.

Response: The authors in reference 1 were corrected in the revised manuscript.

3. Page:3, Line:56-57. ….flavonoids [3-7]; ….products [8-10]. Here over-citation may be avoided.

Response: The authors accepted the advice and the references cited in the two sentences were cut down in the revised manuscript.

4. Page:3, Line:59. ‘Sea buckthorn oil’ instead of ‘sea buckthorn oils’

Response: The phrase was corrected in the revised manuscript.

5. Page: 4, Line 74. Add a reference to the statement. The plant is able to avoid cold and is not resistant, because the leaves are shed under extreme cold condition in this plant. Even the species is not resistant to alkali too.

Response: The authors agreed this opinion and this sentence was revised to ‘Sea buckthorn adapts well to extreme conditions, including drought, salinity, alkalinity, and temperatures [12]’ in the revised manuscript.

12. Ruan CJ, Li H, Mopper S. Characterization and identification of ISSR markers associated with resistance to dried-shrink disease in sea buckthorn. Mol. Breeding. 2009; 24:255−268.

6. Page:4. Line:85. Use full form at first place ‘MAB’.

Response: The sentence was corrected in the revised manuscript and the full form ‘molecular marker-assisted breeding’ was used at first place ‘MAB’. 

7. Page:5. Line:110. What was the premise of including two known elite varieties in the study? Any supportive reference(s) for the statement, and also mention the context in which these varieties are elite.

Response: The premise of elite varieties include high yield, good agronomic traits and strong adaptability to environment, etc. Some Chinese references support that Quyisike and Zhongguoshajiwild are elite cultivars. The word ‘elite’ in the sentence was deleted in the revised manuscript for no English reference supported it.

8. Page:12. Line:204-205. May be included in Material and Methods.

Response: The authors accepted the advice and the sentence ‘Minimum, maximum, mean, standard deviation (SD), and coefficient of variation (CV%) were recorded.’ was added in Material and Methods of the revised manuscript.

Reviewer #2: 

1. The authors mention that 76 varieties were used. There is no mention of the different species they belonged to in M&M, although it has been mentioned later in the text and table. Incorporate that information in the M&M.

Response: The good advice mentioned above is accepted by the authors. The related information has been added in M&M of the revised manuscript. 

2. Are these 76 different varieties or just different accessions? At many places they are being referred to as ‘cultivars’ also. Please correct accordingly in the text wherever mentioned.

Response: After careful consideration, the authors thought ‘accessions’ would be appropriate. The 'varieties' has been replaced into ‘accessions’ in the revised manuscript. 

3. How variable are the climatic conditions of the three research institutes?

Response: The climatic conditions of different growth sites of sea buckthorn samples has been added in S2 Table of the revised manuscript, with the caption ‘Geographical and climatic conditions at different sample collection sites of sea buckthorn in northern China’. 

4. Line 109: ‘………provided 76 varieties’. Does this mean that all the 76 were grown at all the 3 fields? There is no clarity on this aspect in the M&M. Most quantitative traits exhibit a huge variation across environments. To study the phenotypic variations it would have been much informative if all the 76 varieties were grown together across all the three fields. Why was that not considered?

Response: Among the 76 accessions of sea buckthorn samples, 12 were grown in the Institute of Selection and Breeding of Hippophae, 52 were grown in the Research Institute of Berry and 12 were grown in the Jiuchenggong Breeding Base of Sea Buckthorn. These accessions are able to adapt to local climate and screened to be excellent germplasm.

 The authors agreed the opinion that most quantitative traits exhibit a huge variation across environments. We did the comparative analysis on fruit morphological traits of the same cultivars grown in different cultivated fields in our early studies and the data was complemented in the results (S4 Table) of the revised manuscript. The aim in this study is to further screen the elite accessions from the 78 accession with good adaption to the environments of cultivated fields and prepare for the next step of MAB. In the follow-up study, the continuous observation of the environmental factors would be considered. 

5. There is no mention of how these varieties were grown in the field, and data from how many plants were considered for the morphological and oil analysis. For eg. for hundred berry weight (HBW), berries were collected from how many different plants?

Response: The information has been supplemented in the introduction of the revised manuscript. The sea buckthorn samples in this study are collected from 235 individuals (2−5 ramet plants each accession) of 5−8 years in 5 growth sites. The berries of each accession were pooled and frozen as quickly as possible at −20 °C. When all plant materials were harvested, the berries were transferred to –50 °C for storage until analysis. The related information has been supplemented in the M&M and Table 1.

6. Line 137: For the oil extraction and FA analysis, the authors mention that ‘each sample was analyzed three times’. Why weren’t three biological replicates taken for this analysis? 

Response: The authors are sorry for the incorrect expression. In this study, three biological replicates were taken for every analysis. The sentence is corrected in the revised manuscript.

7. Line 180-181: The authors have used 17 previously developed SSR markers and 3 newly developed SSR markers using RNA-Seq. What was the basis of selection of just 3 new markers from the RNA-Seq. Why weren’t more markers deployed for the genetic characterization?

Response: The authors obtained many SSR sequences using RNA-Seq method and designed the primers to screen those SSR loci with polymorphism in sea buckthorn cultivars. We reported 17 developed SSR markers at first. In subsequent experiments, we screened 3 new SSR markers which also showed polymorphic amplification in sea buckthorn germplasm. RNA-Seq SSR loci with polymorphism in sea buckthorn germplasm were difficult to develop for that SSR markers derived from expressed region of genome showed high conservation to some extent in our study. That’s why no more markers deployed for the genetic characterization in sea buckthorn at now.

8. Line 180: Please reframe the sentence. It appears that the authors have done RNA-seq to generate the 3 new SSR markers. Although, the RNA-Seq had been done in previous study from where the 17 SSR were also developed (Reference 17).

Response: The authors accepted the advice and the sentence has been changed into ‘The Twenty polymorphic microsatellite loci (SSR) developed using RNA-Seq were evaluated and loci SB1-SB17 were previously published [17]’ in the revised manuscript.

9. Instead of ‘different origins’ that has been used repeatedly by authors throughout the text and tables, I suggest use the two different species and hybrid accessions.

Response: The authors accepted the good advice. Some ‘different origins’ were changed into the ‘two different subspecies and hybrid accessions’ and the others were deleted in the revised manuscript. 

10. Line 340: ‘All the primers’. Reframe this line. All primers did not give 59 bands. A total of 59 bands were amplified.

Response: The sentence has been revised according to the advice in the revised manuscript. 

11. Line 341: ‘accounting for 86.44%’ . Incomplete sentence, 86.44% of what??

Response: The sentence has been revised according to the advice in the revised manuscript.

12. Line 372: the 3 subgroups have been referred incorrectly. They are IIa, IIb and IIc.

Response: The names of 3 subgroups were corrected in the revised manuscript.

13. Line 421: ‘in comparison of populations’. Statement not clear. Please reframe.

Response: The phrase has been changed to ‘in population identification’ in the revised manuscript.

14. Line 436: ‘gene sequences’. Are all the SSR markers used genic in nature?

Response: SSR can be divided into genomic SSRs and genic SSRs because of the resource of sequences used for SSR identification. Genic SSRs derived from transcriptome or expressed sequence tag sequences are located in expressed genes. These markers can be linked with important phenotypic characteristics through quantitative trait loci analysis. In this study, all SSR markers are genic SSRs.

15. Table 1: Could just be described as the ‘Accessions of sea buckthorn used for the study’

Response: The authors accepted the good advice and the title of Table 1 was revised into the ‘Accessions of sea buckthorn used for the study’.

---

## [Decision Letter · Decision Letter 1]

27 Dec 2019

PONE-D-19-17567R1

Diversity in sea buckthorn (Hippophae rhamnoides L.) accessions with different origins based on morphological characteristics, oil traits, and microsatellite markers

PLOS ONE

Dear Dr. Ruan,

Thank you for submitting your manuscript to PLOS ONE. After careful consideration, we feel that it has merit but does not fully meet PLOS ONE’s publication criteria as it currently stands. Therefore, we invite you to submit a revised version of the manuscript that addresses the points raised during the review process.

We would appreciate receiving your revised manuscript by Feb 10 2020 11:59PM. To enhance the reproducibility of your results, we recommend that if applicable you deposit your laboratory protocols in protocols.io, where a protocol can be assigned its own identifier (DOI) such that it can be cited independently in the future. For instructions see: http://journals.plos.org/plosone/s/submission-guidelines#loc-laboratory-protocols

We look forward to receiving your revised manuscript.

Kind regards,

Shailendra Goel, Ph.D.

Academic Editor

PLOS ONE

Additional Editor Comments (if provided):

Both the reviews for revision have been received. One of the reviewer is mostly satisfied with the response from authors but other still has certain queries. Both the reviewers have pointed out queries regarding presentation of data, especially the inconsistencies in the SSR data. One of the reviewer has provided a sanitised version of MS. The biggest problem Is the quality of written English. The MS requires substantial improvement in English quality. I suggest authors have to take help from a professional. They also have to address the inconsistencies in MS. Authors also need to answer all the questions raised about data.

Reviewers' comments:

Reviewer's Responses to Questions

**Comments to the Author**

1. If the authors have adequately addressed your comments raised in a previous round of review and you feel that this manuscript is now acceptable for publication, you may indicate that here to bypass the “Comments to the Author” section, enter your conflict of interest statement in the “Confidential to Editor” section, and submit your "Accept" recommendation.

Reviewer #1: (No Response)

Reviewer #2: (No Response)

2. Is the manuscript technically sound, and do the data support the conclusions?

Reviewer #1: Yes

Reviewer #2: Yes

3. Has the statistical analysis been performed appropriately and rigorously? 

Reviewer #1: Yes

Reviewer #2: Yes

4. Have the authors made all data underlying the findings in their manuscript fully available?

Reviewer #1: Yes

Reviewer #2: No

5. Is the manuscript presented in an intelligible fashion and written in standard English?

Reviewer #1: Yes

Reviewer #2: No

6. Review Comments to the Author

Reviewer #1: Most of the queries have been answered and incorporated in text appropriately. However, some of them need quick attention.

1. In the abstract authors have mentioned they have used 23 SSR markers out of which 17 were previously reported. Did they assessed 6 new SSRs while in response/text the number of new makers tested is 3. Query:9 (revision 1)

2. The query 7 related to seed width and seed thickness is not answered suitably, though reference is given. Seed thickness can not be regarded as seed height. Seed thickness may be similar to seed width/diameter.

3. Answer/justification for query 12 need to be recheck and if so…..it must be mentioned in materials and methods of the manuscript.

Reviewer #2: General Comments:

The manuscript has a lot of issues with the language. Many such sentences have been highlighted in the document attached. Some of these appear as two half statements fused. At other places, the sentences lack clarity. The authors need to re-frame all such statements.

The Materials & Methods needs to be revised at places (please see suggestions). Most trait measures in various tables lack the unit of measurements. Please incorporate that.

Specific comments:

1. The text inconsistently mentions the deployment of 20 SSR primer pairs at certain places (line 357, 450, Header of Table 5) and at other places (Line 27, 41, 46, 197, 358, 373, 383, 451, 482) the use of 23 SSR primer pairs has been mentioned. The supplementary table (S4) gives sequence information for 23, while its header says 20 SSR primers. Table 5, gives information for 23 markers although the Header says 20. Please ensure that all these ambiguities are taken care of.

2. Abstract says 69 polymorphic bands, while in results 59 polymorphic bands are mentioned. This ambiguity also needs to be addressed.

Introduction:

1. Line 62: ‘Two important parameters in……..oil quantity are oil content’. Oil content cannot be a parameter of oil quality. So, this statement needs modification.

2. Line 78: ‘Due to small berries………………….artificial hybridization for elite accessions.’ The statement needs to be reframed.

3. Line 98: ‘The diversity analysis…………………………germplasm’. The authors appear to have have fused to incomplete sentences. This needs to be re-wrtitten.

All other such statements have been highlighted in the document attached.

Materials & Methods

1. Line 139: ‘There were three biological replicates………measurement’.

Do the authors mean that 300 berries were taken for the analysis? 100 berries from 2-5 plants/ accession is a good enough number for the analysis.

2. Line142: ‘…with over 20 measurements…for each’

This is not clear. Do the authors mean 20 berries per accession?? And how many plants did these berries belong to?

3. For the oil extraction, were the seeds and fruit pulp weighed prior to oil extraction to maintain some uniformity. This has not been mentioned in the M&M.

The oil contents in both seeds and fruit pulp as mentioned in Line 153 is expressed as percentage. Percentage of what? Seed/pulp weight? The authors need to clearly mention that in the M&M.

In the results (Line 278), the authors mention ‘….highest oil content (24.68%) based on dry weight.’ This means that the weight of the pulp/seed was considered. But, this has not been clearly mentioned either in the M&M or in the Table 3. The units for oil characteristic (min and max) have not been mentioned in the table.

4. Line 197: ‘Twenty-three polymorphic microsatellite loci (SSR) developed using RNA-Seq was evaluated and loci SB1-SB17 were previously reported’.

Please mention here the names of the SSR markers (SB1-SB23). Nowhere in the text have they been mentioned except for tables. Then it can be mentioned that SB1-17 were previously deployed (Ref. 14).

The authors need to clearly mention in the introduction itself that in a previous study, RNA seq analysis was done to generate SSR markers and these were tested on 31 accessions. The 17 SSR markers developed in that study have been utilized in the present endeavor for genetic diversity assessment of larger set of accessions. This description in the ‘introduction’ will bring more clarity in the text. This previous study and its outcomes should be mentioned clearly in the ‘Introduction’ so that its extension in the present study can be deciphered.

Results:

1. Line 246: ‘In previous mutilocation trials in Suiling (47°14′N, 127°06′E; 202 m) and Dengkou

(40°43′N, 106°30′E; 1053m, Inner Mongolia), the fruit characteristics of 11 large……’.

How many berries per accession were taken for this analysis? The data should be represented as + SD in Table S7.

2. Line 302: ‘Small variations were found in the proportion of linoleic acid in seed oil (40.44 – 42.87%). Its proportion in hybrids were slightly higher than in ssp. mongolica (42.87% vs 42.10%.....’

Are these differences significant?

3. Table 4: How is the oil content being measured? Total oil per gram weight of seeds and pulp or some other measure?

Tables and Figures

1. Table 1: Since the authors have already mentioned that 2-5 ramet plants were collected per accession. The columns indicating the number of plants taken per accession can be removed from the Table.

2. Table 3: The units for the min. and max values of the oil characteristics have not been mentioned in the table. Similarly mention the units of measurement for each of the component in Table 4.

3. Table 3 & 4: The different fatty acid names should be included in the first column. Example: Oleic (18:1), Palmitic acid (16:0) etc.

4. Table S1: This table again classifies all the lines used as ‘cultivars’. Are these accessions or cultivars? Please check.

5. Table S3 carries a different header than the one that has been listed at the end of the manuscript. Please change that.

6. Table S7: The header for this table has been titled as Table S5. Please correct. Also it mentions ‘two experimental fields’ although it has data from three places. So, please correct.

7. PLOS authors have the option to publish the peer review history of their article (what does this mean?). If published, this will include your full peer review and any attached files.

Reviewer #1: No

Reviewer #2: No

---

## [Author Response · Author response to Decision Letter 1]

24 Jan 2020

General Comments:

The manuscript has a lot of issues with the language. Many such sentences have been highlighted in the document attached. Some of these appear as two half statements fused. At other places, the sentences lack clarity. The authors need to re-frame all such statements.

Response: Thank you for the valuable suggestion. The sentences mentioned above are all re-framed with clarity. And the revised manuscript was professionally edited by American Journal Experts (AJE, ID: HS1GCXH7) for the improvement in English quality.

The Materials & Methods needs to be revised at places (please see suggestions). Most trait measures in various tables lack the unit of measurements. Please incorporate that. 

Response: Thank you for the valuable suggestion. The places mentioned above in the Materials & Methods have been revised according to the reviewers’ suggestions. And the unit of measurements has been added in the tables of the revised manuscript.

Specific comments: 

1. The text contradictorily mentions the deployment of 20 SSR primer pairs at certain places (line 357, 450, Header of Table 5) and at other places (Line 27, 41, 46, 197, 358, 373, 383, 451, 482) the use of 23 SSR primer pairs has been mentioned. The supplementary table (S4) gives sequence information for 23, while its header says 20 SSR primers. Table 5, gives information for 23 markers although the Header says 20. Please ensure that all these ambiguities are taken care of.

Response: Twenty-three SSR primer pairs were used in this study. The number ‘20’ at certain places (line 357, 450, Headers of Table 5 and S4 Table) have been replaced by 23 in the revised manuscript.

2. Abstract says 69 polymorphic bands, while in results 59 polymorphic bands are mentioned. This ambiguity also needs to be addressed.

Response: The number of polymorphic bands is 59. It has been revised in the abstract of the revised manuscript.

Introduction:

1. Line 62: ‘Two important parameters in……..oil quantity are oil content’. Oil content cannot be a parameter of oil quality. So, this statement needs modification.

Response: The authors agreed with this opinion. The oil content is a parameter of oil yield. The sentence has been revised as bellow. 

‘Two important parameters in analyzing oil yield and quality are oil content and fatty acid (FA) composition (referred to here as ‘oil traits’ for simplicity).’

2. Line 78: ‘Due to small berries………………….artificial hybridization for elite accessions.’ The statement needs to be reframed.

Response: Thank you for the valuable suggestion. The statement is reframed as bellow.

‘Due to the small berries and thorns of native cultivars (ssp. sinensis), which result in little economic value, the breeding of sea buckthorn has undergone different stages of development in China, such as introduction, domestication, seedling selection and artificial hybridization for elite accessions.’

3. Line 98: ‘The diversity analysis…………………………germplasm’. The authors appear to have fused to incomplete sentences. This needs to be re-written.

All other such statements have been highlighted in the document attached.

Response: Thank you for the valuable suggestion. All such statements are re-written in the revised manuscript. 

 Materials & Methods

1. Line 139: ‘There were three biological replicates………measurement’.

Do the authors mean that 300 berries were taken for the analysis? 100 berries from 2-5 plants/ accession is a good enough number for the analysis.

Response: Yes. 300 berries were taken for the analysis. We collected more than 300 berries per accession and the analyses of other nutrients were performed in our research work, e.g. vitamin C, vitamin E and carotenoids.

2. Line142: ‘…with over 20 measurements…for each’

This is not clear. Do the authors mean 20 berries per accession?? And how many plants did these berries belong to? 

Response: The authors agreed with this view. It means averaged 20 determinations were done for each character. These berries were selected from the berry samples randomly collected from 2-5 ramet plants per accession. This sentence mentioned above has been re-framed for clarity in the revised manuscript as bellow.

‘The transverse and longitudinal diameters of berries (BTD and BLD) and the length, width and thickness of seeds (SL, SW and ST) were measured over 20 times each (on average) by micrometer calipers.’

3. For the oil extraction, were the seeds and fruit pulp weighed prior to oil extraction to maintain some uniformity. This has not been mentioned in the M&M. 

The oil contents in both seeds and fruit pulp as mentioned in Line 153 is expressed as percentage. Percentage of what? Seed/pulp weight? The authors need to clearly mention that in the M&M.

In the results (Line 278), the authors mention ‘….highest oil content (24.68%) based on dry weight.’ This means that the weight of the pulp/seed was considered. But, this has not been clearly mentioned either in the M&M or in the Table 3. The units for oil characteristic (min and max) have not been mentioned in the table.

Reponse: Thank you for the valuable suggestion. The method of lipid extraction was described by Yang and Kallio (2001). Samples (1 g) of seeds and fruit pulp were isolated from freeze-dried berries and lipids from the samples were extracted with chloroform/methanol (2:1, v/v) with mechanical homogenization of the tissues. The oil contents (percentages) in seeds and fruit pulp were calculated (oil % in seeds and lyophilized fruit pulp). The fatty acid composition was also expressed as a weight percentage of the total fatty acids. The units (weight percentages) for oil characteristics in Table 3 and Table 4 have been added in the revised manuscript.

4. Line 197: ‘Twenty-three polymorphic microsatellite loci (SSR) developed using RNA-Seq was evaluated and loci SB1-SB17 were previously reported’. 

Please mention here the names of the SSR markers (SB1-SB23). Nowhere in the text have they been mentioned except for tables. Then it can be mentioned that SB1-17 were previously deployed (Ref. 14).

The authors need to clearly mention in the introduction itself that in a previous study, RNA seq analysis was done to generate SSR markers and these were tested on 31 accessions. The 17 SSR markers developed in that study have been utilized in the present endeavor for genetic diversity assessment of larger set of accessions. This description in the ‘introduction’ will bring more clarity in the text. This previous study and its outcomes should be mentioned clearly in the ‘Introduction’ so that its extension in the present study can be deciphered.

Reponse: Thank you for the valuable suggestion. The sentences mentioned above have been re-framed for clarity in the revised manuscript as bellow. 

‘Twenty-three polymorphic microsatellite loci (SB1-SB23) developed using RNA-Seq were evaluated. Of these, 17 (SB1-SB17) had been deployed in a previous study by the group [14].’

And the authors added the statements of 17 RNA-Seq SSR markers developed in our previous study and mentioned these SSR markers have been utilized in the present endeavor for genetic diversity assessment of larger set of accessions in the revised manuscript.

‘In our previous study, 17 RNA-Seq SSR markers (SB1-SB17) were developed and validated on 31 accessions, which were utilized in the present study for genetic diversity assessment of larger set of accessions [14].’

Results

1. Line 246: ‘In previous mutilocation trials in Suiling (47°14′N, 127°06′E; 202 m) and Dengkou (40°43′N, 106°30′E; 1053m, Inner Mongolia), the fruit characteristics of 11 large……’. 

How many berries per accession were taken for this analysis? The data should be represented as + SD in Table S7. 

Reponse: 300 berries of each cultivar were randomly sampled and divided into 3 groups (100 berries were divided into 1 group) to determine the hundred berry weight (HBW). 20 berries of each cultivar were randomly sampled to determine the transverse, longitudinal diameters of berries and berry shape indices (BTD, BLD and BSI). The data has been represented as + SD in S7 Table in the revised manuscript.

2. Line 302: ‘Small variations were found in the proportion of linoleic acid in seed oil (40.44 – 42.87%). Its proportion in hybrids were slightly higher than in ssp. mongolica (42.87% vs 42.10%.....’ 

Are these differences significant?

 Reponse: These differences are significant despite small variations. The content of seed oil in hybrids is lower than that in ssp. mongolica. However, the proportion of linoleic acid (an important polyunsaturated fatty acid) in seed oil is higher in hybrids than that in ssp. mongolica, which showed high oil quality of seed oil in hybrids.

3. Table 4: How is the oil content being measured? Total oil per gram weight of seeds and pulp or some other measure? 

Reponse: The method of lipid extraction was described by Yang and Kallio (2001). Samples (1 g) of seeds and fruit pulp were isolated from freeze-dried berries and lipids from the samples were extracted with chloroform/methanol (2:1, v/v) with mechanical homogenization of the tissues. The oil contents (percentages) in seeds and fruit pulp were calculated (oil % in seeds and lyophilized fruit pulp). 

Tables and Figures

1. Table 1: Since the authors have already mentioned that 2-5 ramet plants were collected per accession. The columns indicating the number of plants taken per accession can be removed from the Table.

Reponse: The columns indicating the number of plants taken per accession have been removed from Table 1 in the revised manuscript.

2. Table 3: The units for the min. and max values of the oil characteristics have not been mentioned in the table. Similarly mention the units of measurement for each of the component in Table 4.

Reponse: The units of the oil characteristics have been added in the headers of Table 3 and Table 4.

3. Table 3 & 4: The different fatty acid names should be included in the first column. Example: Oleic (18:1), Palmitic acid (16:0) etc.

Reponse: Thank you for the valuable suggestion. The different fatty acid names are included in the first column of Table 3 and Table 4 according to the examples.

4. Table S1: This table again classifies all the lines used as ‘cultivars’. Are these accessions or cultivars? Please check.

Reponse: The ‘accession’ has replaced the ‘cultivar’ in S1 Table of revised manuscript.

5. Table S3 carries a different header than the one that has been listed at the end of the manuscript. Please change that.

Reponse: The header of S3 Table listed at the end of the manuscript has been changed in the revised manuscript.

6. Table S7: The header for this table has been titled as Table S5. Please correct. Also it mentions ‘two experimental fields’ although it has data from three places. So, please correct.

Reponse: The header for S7 Table has been corrected. The mutilocation trials were performed in two experimental fields (Suiling and Dengkou). Russia is the country of origin of those cultivars and the related data were provided by the units where they were introduced.

---

## [Decision Letter · Decision Letter 2]

14 Feb 2020

PONE-D-19-17567R2

Diversity in sea buckthorn (Hippophae rhamnoides L.) accessions with different origins based on morphological characteristics, oil traits, and microsatellite markers

PLOS ONE

Dear Dr. Ruan,

Thank you for submitting your manuscript to PLOS ONE. After careful consideration, we feel that it has merit but does not fully meet PLOS ONE’s publication criteria as it currently stands. Therefore, we invite you to submit a revised version of the manuscript that addresses the points raised during the review process.

ACADEMIC EDITOR: 

Both the reviewers have agreed with the changes made in Manuscript. One of the reviewer has suggested some changes in the MS and has provided a sanitized version of MS, the authors are requested to include these changes in MS. I think we will not require any further reviewing and as and when MS is submitted with required changes, the MS can be accepted. 

We would appreciate receiving your revised manuscript by Mar 30 2020 11:59PM. To enhance the reproducibility of your results, we recommend that if applicable you deposit your laboratory protocols in protocols.io, where a protocol can be assigned its own identifier (DOI) such that it can be cited independently in the future. For instructions see: http://journals.plos.org/plosone/s/submission-guidelines#loc-laboratory-protocols

We look forward to receiving your revised manuscript.

Kind regards,

Shailendra Goel, Ph.D.

Academic Editor

PLOS ONE

Reviewers' comments:

Reviewer's Responses to Questions

**Comments to the Author**

1. If the authors have adequately addressed your comments raised in a previous round of review and you feel that this manuscript is now acceptable for publication, you may indicate that here to bypass the “Comments to the Author” section, enter your conflict of interest statement in the “Confidential to Editor” section, and submit your "Accept" recommendation.

Reviewer #1: All comments have been addressed

Reviewer #2: All comments have been addressed

2. Is the manuscript technically sound, and do the data support the conclusions?

Reviewer #1: Yes

Reviewer #2: Yes

3. Has the statistical analysis been performed appropriately and rigorously? 

Reviewer #1: Yes

Reviewer #2: Yes

4. Have the authors made all data underlying the findings in their manuscript fully available?

Reviewer #1: Yes

Reviewer #2: Yes

5. Is the manuscript presented in an intelligible fashion and written in standard English?

Reviewer #1: Yes

Reviewer #2: (No Response)

6. Review Comments to the Author

Reviewer #1: As most of my queries have been attended, the MS may accepted for publication provided the other reviewer has also reached a similar decision.

Reviewer #2: The authors have made the suggested changes and the manuscript should be accepted for publication after few minor changes as suggested below. These have also been highlighted in the manuscript attached.

1. Some text changes are suggested: Line 36, 77, 78, 84, 87, 362, 375, 452, 461, 462, 489

2. Line 307, 345: Lacks clarity.

3. Suggest changing Table 5 title to: ‘Genetic diversity analyses of 78 accessions of sea buckthorn germplasm using 23 SSR markers’.

4. As previously suggested to the authors, since for each of the traits multiple measurements were taken, please include +/- SD values for each trait/accession in Table S6.

7. PLOS authors have the option to publish the peer review history of their article (what does this mean?). If published, this will include your full peer review and any attached files.

Reviewer #1: No

Reviewer #2: No

---

## [Author Response · Author response to Decision Letter 2]

25 Feb 2020

The authors thank two reviewers for their careful reading, comments and suggestion. We revised our manuscript in the best way as we could. Revised portions are marked in red in the revised manuscript. For the individual comments see our reply below.

Reviewer #1: As most of my queries have been attended, the MS may accepted for publication provided the other reviewer has also reached a similar decision.

Response: The authors thank for your acceptance of the revised MS.

Reviewer #2: The authors have made the suggested changes and the manuscript should be accepted for publication after few minor changes as suggested below. These have also been highlighted in the manuscript attached.

1. Some text changes are suggested: Line 36, 77, 78, 84, 87, 362, 375, 452, 461, 462, 489.

Response: The authors thank for your suggestions. The text changes mentioned above have been corrected in the revised MS. The details are as follows.

Line 36: The word ‘approximately’ has been deleted.

Line 77,78: The sentence has been revised to ‘However, the fruits of native cultivars are small and thorny and of low economic value, which encourages the breeding of sea buckthorn has undergone different stages of development in China’.

 Line 84: The word ‘a’ has been deleted.

 Line 87: The word ‘associated’ has been deleted.

 Line 362: The word ‘in’ has been revised to ‘among’.

 Line 375: The sentence has been changed into ‘The characteristics of 23 polymorphic SSR markers in sea buckthorn accessions are shown in Table 5’.

 Line 452: The word ‘identify’ has been revised to ‘investigate the genetic relationships among 78 sea buckthorn accessions’.

 Line 461: The phrase ‘between ssp. sinensis and ssp. mongolica accessions’ has been changed into ‘between accessions of ssp. sinensis and ssp. mongolica’.

 Line 462: The words ‘This result illustrated’ has been revised to ‘This uniformity indicated’.

 Line 489: The word ‘Hippophae’ has been revised to ‘H.’.

2. Line 307, 345: Lacks clarity.

Response: The authors accepted the opinion and revised these sentences for clarity. The details are as follows.

Line 307: The sentence ‘and had the highest value of the samples from the two different subspecies and hybrid accessions’ has been revised to ‘and showed the highest mean value among the two different subspecies and hybrid accessions’.

Line 345: The sentence ‘that were primarily provided by a marker of BLD’ has been revised to ‘that were primarily provided by the phenotypic character of BLD’.

3. Suggest changing Table 5 title to: ‘Genetic diversity analyses of 78 accessions of sea buckthorn germplasm using 23 SSR markers’.

Response: The authors thank for your suggestion. The title of Table 5 has been changed into ‘Genetic diversity analyses of 78 accessions of sea buckthorn germplasm using 23 SSR markers’ in the revised MS.

4. As previously suggested to the authors, since for each of the traits multiple measurements were taken, please include +/- SD values for each trait/accession in Table S6.

Response: The authors thank for you suggestion. The SD values for each trait/accession have been added in Table S6.

---

## [Editor Report · Decision Letter 3]

28 Feb 2020

Diversity in sea buckthorn (Hippophae rhamnoides L.) accessions with different origins based on morphological characteristics, oil traits, and microsatellite markers

PONE-D-19-17567R3

Dear Dr. Ruan,

We are pleased to inform you that your manuscript has been judged scientifically suitable for publication and will be formally accepted for publication once it complies with all outstanding technical requirements.

With kind regards,

Shailendra Goel, Ph.D.

Academic Editor

PLOS ONE

Additional Editor Comments (optional):

Both the reviewers are now satisfied by the changes made in the MS. Henceforth, I recommend the publication of this MS in PLOSOne. I thank you for your patience during the process and hope that authors will appreciate all the efforts from the two reviewers in improving the manuscript.
---

## [Editor Report · Acceptance letter]

3 Mar 2020

PONE-D-19-17567R3 

Diversity in sea buckthorn (*Hippophae rhamnoides* L.) accessions with different origins based on morphological characteristics, oil traits, and microsatellite markers 

Dear Dr. Ruan:

I am pleased to inform you that your manuscript has been deemed suitable for publication in PLOS ONE. Congratulations! Your manuscript is now with our production department. 

With kind regards,

on behalf of

Dr. Shailendra Goel 

Academic Editor

PLOS ONE